# A landscape of gene expression regulation for synovium in arthritis

Feng Jiang [1,3], Shou-Ye Hu[2,3], Wen Tian [1,3], Nai-Ning Wang[1], Ning Yang[1], Shan-Shan Dong [1], Hui-Miao Song[1], Da-Jin Zhang[1], Hui-Wu Gao[1], Chen Wang[1], Hao Wu[1], Chang-Yi He[1], Dong-Li Zhu[1], Xiao-Feng Chen [1], Yan Guo [1], Zhi Yang[2] ✉ & Tie-Lin Yang [1] ✉

The synovium is an important component of any synovial joint and is the major target tissue of inflammatory arthritis. However, the multi-omics landscape of synovium required for functional inference is absent from large-scale resources. Here we integrate genomics with transcriptomics and chromatin accessibility features of human synovium in up to 245 arthritic patients, to characterize the landscape of genetic regulation on gene expression and the regulatory mechanisms mediating arthritic diseases predisposition. We identify 4765 independent primary and 616 secondary *cis*-expression quantitative trait loci (*cis*-eQTLs) in the synovium and find that the eQTLs with multiple independent signals have stronger effects and heritability than single independent eQTLs. Integration of genome-wide association studies (GWASs) and eQTLs identifies 84 arthritis related genes, revealing 38 novel genes which have not been reported by previous studies using eQTL data from the GTEx project or immune cells. We further develop a method called eQTac to identify variants that could affect gene expression by affecting chromatin accessibility and identify 1517 regions with potential regulatory function of chromatin accessibility. Altogether, our study provides a comprehensive synovium multi-omics resource for arthritic diseases and gains new insights into the regulation of gene expression.

Synovium, composed of loose connective tissue, is the major site of inflammation in arthritic diseases such as osteoarthritis (OA), rheumatoid arthritis (RA), and juvenile idiopathic arthritis (JIA)[1]. In inflammatory arthritis, activated synovial fibroblasts produce enzymes such as MMPs that degrade joint structures and promote the inflammatory process[2].

Genome-wide association studies (GWASs) have identified hundreds of loci for arthritic diseases[3,4]. Most of these variants are located in noncoding regions, which affect diseases primarily through regulatory mechanisms on the transcriptome. Expression quantitative trait loci (eQTL) have proven useful in addressing the regulatory

mechanisms of GWAS variants[5–7]. However, eQTL signals are generally tissue-specific[6,8,9], and the ability to detect mechanistically informative expression effects depends on assaying expression data from sufficient numbers of samples from disease-relevant tissues[10]. Synovium is an essential tissue associated with many arthritic diseases, but the largest human synovium sequencing dataset published to date includes only 77 samples[11]. Therefore, it is necessary to conduct eQTL analysis for synovium in a relatively large population.

Many eQTLs influence gene expression through effects on chromatin, such as altering regulatory element activity[12–14]. Some studies have analyzed the effect of variants on chromatin state by measuring

[1]Key Laboratory of Biomedical Information Engineering of Ministry of Education, Biomedical Informatics & Genomics Center, School of Life Science and Technology, Xi'an Jiaotong University, Xi'an 710049, P.R. China. [2]Department of Joint Surgery, Honghui Hospital, Xi'an Jiaotong University, Xi'an, Shaanxi 710054, P.R. China. [3]These authors contributed equally: Feng Jiang, Shou-Ye Hu, Wen Tian. ✉e-mail: xgcgfd@126.com; yangtielin@xjtu.edu.cn

chromatin accessibility and gene expression at the population level[13–15]. However, the exploration of this process is limited by the lack of large panels of reference chromatin accessibility features from relevant tissues.

In this work, we identify 5381 independent eQTLs and 4765 eQTL genes based on the genomic and transcriptomic features of human synovium in up to 245 OA patients. With much larger sample size than the previous study (245 vs 77), our work provides an eQTL resource for understanding the role of synovium in arthritis. We also integrate our identified synovium eQTLs with GWAS summary data for multiple arthritis diseases and uncover many novel effect genes which have not been reported before. Lastly, we develop the eQTac method, which could identify variants that could affect gene expression by affecting chromatin accessibility without population-scale ATAC-seq data. Taken together, our work has significant implications for understanding how variants function in synovium and the development of arthritic diseases.

## Results

### Synovium *cis*-eQTL identification and characterization

To identify genetic loci associated with transcript abundance in synovial tissue, we generated gene expression and genotype data from a population of 245 OA patients who underwent knee replacement surgery (Fig. 1a, Supplementary Table 1). After quality control, we obtained 4,260,261 single nucleotide polymorphisms (SNPs) and 19,381 expressed genes from 202 individuals with matched genomes and transcriptomes.

*Cis*-eQTLs were calculated for all expressed genes in the *cis* region (<1 Mb) with additional covariates (sex, age, PEER factors, and genotype PCs) (see "Methods" section). We identified 429,021 SNPs (eSNPs) significantly associated with expressions of 4765 genes (eGenes) at a 5% false discovery rate (FDR). There are 27,013 common eSNPs and 462 common eGenes between our results and the previously published synovium eQTL study in 77 individuals[11], which identified 67,501 eSNPs and 868 eGenes (Supplementary Fig. 1).

Most of the eSNPs were only associated with one eGene (Supplementary Fig. 2). The eSNP hot spot region extended from 100 kb upstream of the transcriptional start site (TSS) to 50 kb downstream of the transcriptional end site (TES). The most significant eSNP for eGene is named lead eSNP. Approximately 38% (1803/4765) and 28% (1339/4765) of lead eSNPs were located upstream and downstream of the target eGenes, respectively (Fig. 1b). Besides the known regulatory regions, lead eSNPs were also located in the intergenic (35%) and intronic regions (42%) (Fig. 1b), supporting the regulatory effects of noncoding regions.

The effect sizes of lead SNPs on target genes were decreased significantly with increasing distance to the target eGene TSS (Pearson $r = −0.14$, $P = 2.84 \times 10^{-23}$, Fig. 1c, Supplementary Fig. 3). As the effect sizes of lead SNPs were positively correlated with the coefficients of variation (CVs) in target gene expressions even after adjusting gene expression levels (Pearson $r = 0.22$, $P = 7.32 \times 10^{-53}$, Fig. 1d and Supplementary Fig. 4), genes located far from their lead SNPs should have small variations among population. However, we found that some genes furthest from their lead SNPs exhibited the largest variation (Fig. 1e, see "Methods" section), indicating the presence of long-range regulatory effects. A similar observation has been made in mouse embryonic stem cells (mESCs), where the gene expression CVs among cells were positively correlated with the enhancer-promoter distance[16].

Our analysis also revealed that eGenes and non-eGenes exhibited different tolerance levels to loss-of-function mutations (Fig. 1f, see "Methods" section). Highly expressed genes without any eQTL effect in synovium were less tolerant to loss-of-function mutations in their coding region than eGenes. This phenomenon has also been observed in blood eQTL[7], suggesting that eQTLs may enhance the tolerance of target genes to loss-of-function mutations.

### EQTLs with multiple independent signals showed stronger effects

To identify the independent signals for eQTLs, we conducted a stepwised conditional analysis and characterized the independent eQTLs (see "Methods" section). We defined the independent primary eQTLs with the highest ranking for each eGene, and the remaining independent signals were referred to as secondary eQTLs. In total, we identified 4765 primary eQTLs and 616 secondary eQTLs (Supplementary Data 1). Of the 4765 eGenes with primary eQTLs, 4201 eGenes (88.2%) contained only one independent eQTLs, while 564 eGenes exhibited significant secondary eQTLs (Fig. 2a), and most of the secondary eQTLs were in close proximity to the primary eQTLs (Fig. 2b). Specifically, the secondary eQTLs were located further away from the TSS than primary eQTLs (median = 34.7 kb vs 23.9 kb, two-sided Mann-Whitney test, $P = 1.12 \times 10^{-6}$) (Fig. 2c). This pattern might be primarily caused by eQTLs located outside the genes. The distance showed no difference whether or not normalized by gene length for independent eQTLs located on genes (Supplementary Fig. 5).

For eGenes with multiple independent eQTLs, the effects of primary eQTLs were stronger than single signal primary eQTLs (Fig. 2d), and explained higher ratio of gene expression variance (Fig. 2e, see "Methods" section). Furthermore, eGenes with multiple independent eQTLs showed higher heritability (Fig. 2f, see "Methods" section) and higher tolerance to loss-of-function mutations (Fig. 2g). Together, these findings suggest that characterizing independent signals rather than just the top SNP for each eQTL, could enhance the understanding of gene regulation.

### Functional properties of independent eQTLs

To assess the functional properties of independent eQTLs, we conducted the assay for transposase-accessible chromatin with high-throughput sequencing (ATAC-seq) and identified 154,649 accessible chromatin peaks in synovium of 10 arthritis patients (see "Methods" section, Supplementary Table 1). ATAC-seq signals of independent eQTLs were significantly higher than background SNPs and adjacent regions (Supplementary Fig. 6), and the independent eQTLs showed significant enrichment in open chromatin regions (Fisher exact test, $P = 1.58 \times 10^{-48}$, fold change = 2.98, Fig. 3a). Using public histone modification data in synovium (Supplementary Fig. 7, Supplementary Table 2), we also observed significant enrichment of eQTLs in active histone markers (H3K27ac, H3K4me1, H3K4me3, H3K36me3), and a concomitant depletion in heterochromatin marker (H3K9me3). We have also compared our results with previous eQTL studies in other tissues. Similarly, eQTL SNPs are enriched in the regions of transcriptionally active chromatin (e.g., H3K4me3, H3K27ac) and depleted in the heterochromatin (e.g., H3K9me3)[17,18]. These results suggest that most *cis*-eQTL effects might be driven by genetic perturbations in regulatory elements in active chromatin regions. Moreover, independent eQTLs were positively enriched in 84% (105/125, FDR < 0.05) transcription factors binding sites (TFBSs) from the GTRD database[19] in ATAC peaks (Fig. 3b), indicating the importance of TF binding in the regulation of eQTL variants and open chromatin. The top 10 TFs are enriched in immune-related gene ontology (GO) terms (Fig. 3c, Supplementary Table 3). Specifically, 7 of the 10 TFs have been reported to be involved in the immune cell differentiation or immune response process (Supplementary Table 3).

### Tissue specificity of independent eQTLs

To explore the tissue specificity of synovium, we compared synovium-independent eQTLs with 49 other tissues from the GTEx project[6]. There were 41.6% of the independent eQTLs showed significant signals (significant SNP-gene pairs, FDR < 0.05) in synovium, but not in any GTEx tissues. We further performed mashR analysis[20] (see "Methods" section) to estimate the tissue specificity more stringently. The results showed that the proportion of synovium eQTLs not shared with other

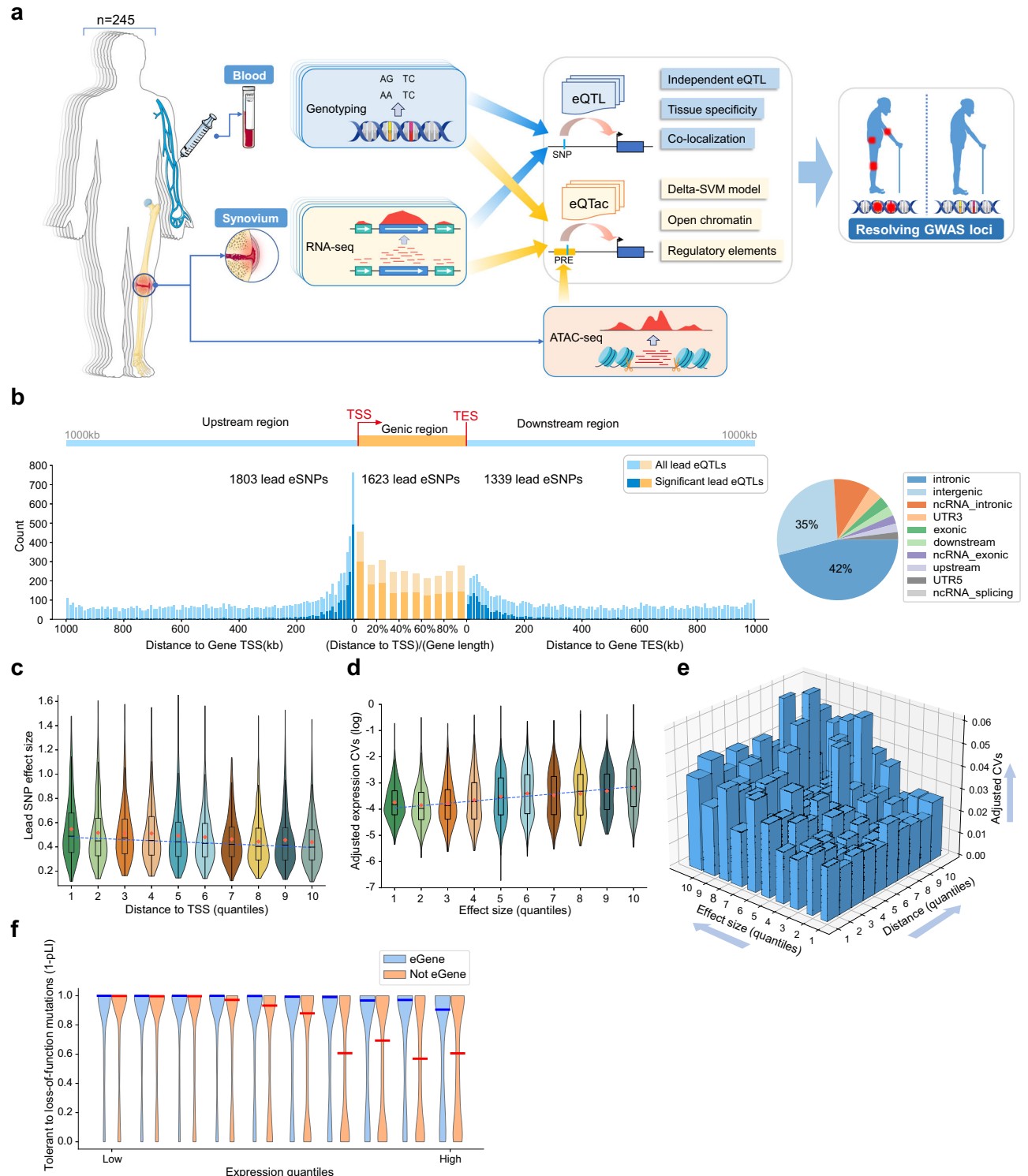

tissues was ranged from 0.19 to 0.53 (Fig. 3d, Supplementary Fig. 8), indicating the presence of tissue-specific genetic regulation in synovium. In addition, we compared our synovium eQTL data with immune cell eQTLs from DICE[21] and BLUEPRINT[22]. The results (Supplementary Fig. 9) showed that our synovium shared over 40% with stimulated CD4+ and CD8+ T cells, NK cells and B cells, supporting the immune signatures we observed from the top enriched TFs (Fig. 3b, c). The shared score was higher than that obtained using the previous synovium eQTLs data[11] (Supplementary Fig. 9).

Considering the genomic position of synovium-specificity eQTLs, we found that local eQTLs are more likely to be shared among different

tissues, while distal eQTLs showed more synovium-specific manner (Fig. 3e). It may be because tissue-specific eQTLs are more likely affected by tissue-specific factors such as distal regulatory elements.

## Colocalization of eQTLs and synovium-related GWAS Loci

To investigate the relationships between synovium-specific regulation of gene expression and arthritic diseases predisposition alleles, we used our eQTL data to annotate disease variants related to synovium in 25 GWAS datasets (Supplementary Data 2). We identified significant enrichment of heritability for eQTLs at OA and RA GWAS loci using partitioned heritability analysis[23] (Fig. 4a, see "Methods" section). In

**Fig. 1 | Overview of synovium eQTLs. a** Overview of the study design. eQTL, expression quantitative trait loci; PRE, potential regulatory element; eQTac, expression quantitative trait accessible chromatin. **b** Distribution of all lead SNPs and significant (FDR < 0.05) lead eSNPs relative to the target genes body. The position of upstream SNPs was relative to target gene TSS, the position of downstream SNPs was relative to target gene TES, and the position of SNPs located on the target gene was (distance to TSS) / (target gene length). Pie plot showed functional annotation for all lead eSNPs by ANNOVAR. **c** The lead SNPs effect sizes in different distances to target gene TSS. Distances to TSS were split into ten intervals according to the quantiles, larger quantile number indicates larger distance values. $N = 4765$ significant lead eQTLs. **d** The target gene expression coefficient of variations (CVs) in different lead SNP effect sizes. Lead SNP effect sizes were split into ten intervals according to the quantiles, larger quantile number indicates larger effect size values. Gene expression values were used as covariates for CVs.

$N = 4765$ significant lead eQTLs. The boxplots in (**c**) and (**d**) represent 25th, 50th, and 75th percentiles, and whiskers extend to 1.5 times the interquartile range. Orange rhombuses represent the mean of each box. The fitting line to the median values is shown in blue. **e** The three-dimensional bar graph showed the target gene expression CVs in different lead SNPs effect sizes and different distances to target gene TSS, larger quantile number indicates larger distance/effect size values. The widths of the bar represent the count of genes in this interval. Gene expression values were used as covariates for CVs. **f** The tolerance to loss-of-function mutations of eGene and non-eGene in different expression levels. The tolerance was denoted by 1-pLI, in which pLI represent the probability of being loss-of-function (LoF) intolerant. The red line and blue line represent the median of each box. Panel **a** was partly generated using Servier Medical Art, provided by Servier, licensed under a Creative Commons Attribution 3.0 Unported License.

addition, we used mental disorder GWASs as negative control, which should not share synovium-specific disease mechanisms. Indeed, there was no enrichment between synovium eQTLs and mental disorder GWAS loci.

To further identify effector genes driving GWAS signals, we conducted colocalization analysis in the 25 synovium-related GWAS summary datasets, to ascertain whether the same variant was associated with both disease and gene expression levels (see "Methods" section). We detected evidence of colocalization for 42 genes in RA, 34 genes in OA, 10 genes in ankylosing spondylitis (AS), and 1 gene in juvenile idiopathic arthritis (JIA), respectively (Fig. 4b, Supplementary Data 3). We compared our results with previous studies using eQTL data from the GTEx project[24–28] or immune cells[28–32] to identify colocalized genes (Supplementary Table 4). As shown in Fig. 4b, the numbers of novel genes specifically identified by our synovium dataset for OA, RA, and AS were 18, 18, and 2, respectively. As for JIA, the only one colocalized gene was also identified by previous studies. We further constructed a protein–protein interaction (PPI) network for the 84 colocalized genes using the STRING database[33]. As shown in Supplementary Fig. 10, the hub genes with the highest degree in the network (*CCR6*, *CD40*, *IRF5*, *ERBB2*, and *LRRK2*) mainly participate in the immune-related process. *CCR6*, *IRF5*, and *CD40* are all well-known autoimmune disease genes associated with RA or AS[34–37]. Their eQTL signals were colocalized with RA GWAS in high posterior probability (Supplementary Fig. 11). Specifically, *ERBB2* and *LRRK2* are firstly identified by our synovium dataset as colocalized genes for RA and AS, respectively. *ERBB2* is a known oncogene with significant role in mediating tumor immune response[38,39]. *LRRK2* is highly expressed in immune cells[40–42]. Mutant *LRRK2* could exacerbate immune response and neurodegeneration in a chronic model of experimental colitis[43].

Moreover, 35.7% (30/84) of the colocalized genes showed differential expression between normal synovium and arthritis synovium (Supplementary Data 3, Supplementary Table 5). Gene ontology (GO) and disease ontology (DO) enrichment analysis showed that the genes colocalized with RA are primarily involved in immune-related GO or DO terms, particularly the immune cellular function such as B cell, T cell, leukocyte, lymphocyte, and some autoimmune diseases (Supplementary Fig. 12, and Supplementary Data 4). For the other three diseases (OA, JIA, and AS), no significant GO/DO enrichment result was obtained since the number of their colocalized genes was limited.

To validate the effect of SNPs on gene expression and cellular phenotypes, we selected an RA-associated SNP rs142845557 to conduct functional experiment. eQTL analysis showed that the allele A of rs142845557 was significantly associated with increased expression of *JAZF1* ($P = 7.3 \times 10^{-8}$, $\beta = 0.28$, Supplementary Fig. 13) with high probability of colocalization (PP.H4 = 0.92, Fig. 4c). Epigenomic annotation analysis showed that rs142845557 is located in histone markers of active enhancer (H3K27ac and H3k4me1, Supplementary Fig. 14). The rs142845557 was homozygous AA associated with increased *JAZF1* expression in MH7A cells (Supplementary Fig. 15a), hence we deleted a

358-bp genomic region containing rs142845557-AA using CRISPR/Cas9 in MH7A cells (Fig. 4d). The deletion efficiency was confirmed by gel electrophoresis experiments (Supplementary Fig. 15b). As shown in Fig. 4d, significantly decreased *JAZF1* expression ($P < 0.01$) was detected in rs142845557-AA deleted cells (KO) compared with the normal control cells, indicating the regulation role of this SNP on *JAZF1* expression. We further conducted a series of functional experiments to examine the cellular phenotypes in rs142845557-KO cells, including migration, invasion, proliferation, and apoptosis. Compared with the control cells, wound-healing and transwell assays showed that the migration and invasion abilities of MH7A were significantly increased in the KO cells ($P < 0.05$) (Fig. 4e, f). TUNEL apoptosis experiment revealed significantly reduced apoptosis in the KO cells (Fig. 4g). CCK-8 assay showed significantly enhanced cell proliferation ability in the KO cells (Fig. 4h). Taken together, our results reveal the regulatory effect of the SNP rs142845557 on target gene *JAZF1* expression and cellular phenotypes, highlighting the importance of this GWAS SNP involved in the pathogenesis of RA.

## Identification of expression quantitative trait accessible chromatin (eQTac)

Since chromatin accessibility is a key factor influencing gene expression, we want to identify eQTLs that can regulate gene expression by affecting chromatin accessibility. Due to the lack of large panels of reference chromatin accessibility features from synovium, we developed a method to predict the chromatin accessibility features for each individual (see "Methods" section). To achieve this, we trained a gapped k-mer support vector machine (gkm-SVM) model[44,45] on 154,649 open chromatin sequences and negative sequences with matched length, GC content, and repeat fraction (Fig. 5a). We used 3-fold cross-validation to tested different combinations of hyper-parameters, and chose the best-performing hyper-parameter to train the prediction model (Supplementary Fig. 16, see "Methods" section). The best model achieved an area under the receiver operating characteristic curve (AUC) of 0.92. (Supplementary Fig. 17a). We also tested the model in another independent synovium dataset[46], which contains ATAC-seq data from 11 subjects. As shown in Supplementary Fig. 17b, the AUC ranged from 0.82 to 0.85, supporting the robustness of the model.

We defined the potential regulatory elements (PREs) as the ±250 bp region surrounding each peak summit, following the enhancer definition of the Activity by Contact (ABC) model[47]. We also filtered out PREs that contained only one independent SNPs (LD $R^2 = 0.3$) or SNPs less than 10 bp apart, to avoid collinearity or disturbance in prediction. Finally, we obtained 10,241 PREs (Fig. 5a). Then we used the prediction model to calculated chromatin accessibility score for these 10,241 PREs for each individual (Fig. 5a; see "Methods" section). We computed the correlation between accessibility score and expression for each PRE-gene pair to identify the regulatory accessible chromatin, which are named as expression quantitative trait accessible chromatin (eQTac) (see "Methods" section). We validated the performance of the

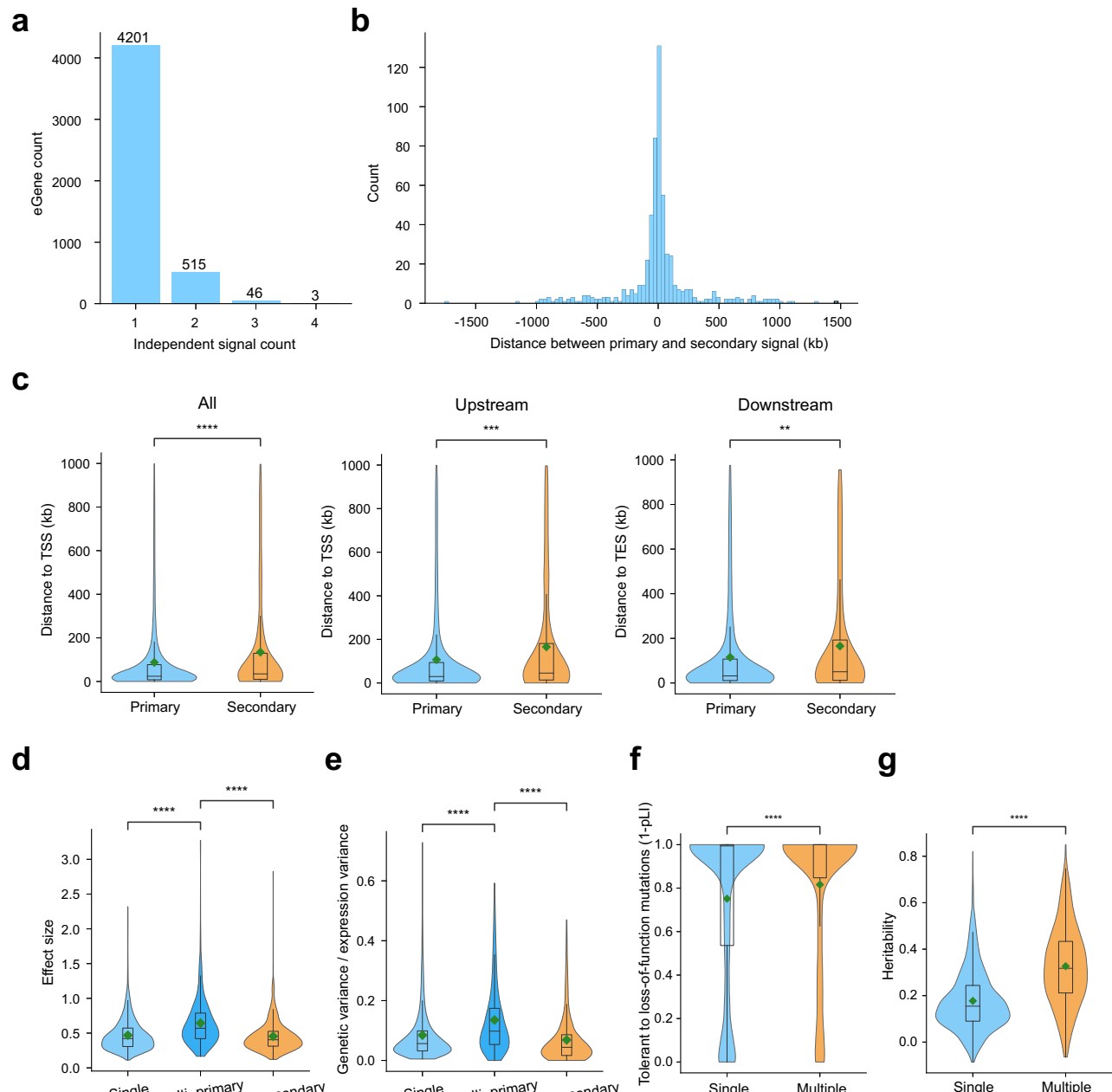

**Fig. 2 | Independent signals of synovium eQTLs. a** The distribution of eGene counts with different counts of independent eQTLs. **b** The distribution of distances between primary eQTLs and secondary eQTLs for eGenes with multiple independent eQTLs (primary signal - secondary signal). **c** The distances to target genes of primary eQTLs and secondary eQTLs in different locations of the gene body. $N = 5380, 2047, 1535$ for all, upstream, and downstream independent eQTLs, respectively. $P = 1.12 \times 10^{-6}, 4.00 \times 10^{-4}, 4.39 \times 10^{-3}$ for all, upstream, and downstream independent eQTLs, respectively. **d** Violin plot showed the effect sizes (absolute value) for three kinds of independent eQTLs. (1) single: the primary eQTL for eGenes with single independent eQTL. (2) multi_primary: the primary eQTL for eGenes with multiple independent eQTLs. (3) multi_secondary: the secondary eQTL for eGenes with multiple independent eQTLs. $P = 4.367 \times 10^{-43}, 8.66 \times 10^{-34}$ for single vs. multi_primary and multi_primary vs. multi_secondary, respectively. Outliers were not shown in the plot. **e** The ratio of explained expression variance for three kinds of independent eQTLs (same as (**d**)). $P = 6.43 \times 10^{-35}, 6.22 \times 10^{-40}$ for single vs. multi_primary and multi_primary vs. multi_secondary, respectively. **f** The heritability of eGenes with single independent eQTLs and multiple independent eQTLs. $P = 3.60 \times 10^{-9}$. **g** The tolerance to loss-of-function mutations of eGenes with single independent eQTLs and multiple independent eQTLs. $P = 4.20 \times 10^{-82}$. The tolerance was denoted by 1-pLI, in which pLI represent the probability of being loss-of-function (LoF) intolerant. $N = 5380$ for independent eQTLs in (**d**)−(**g**). The boxplots in (**c**)−(**g**) represent 25th, 50th (median), and 75th percentiles, and whiskers extend to 1.5 times the interquartile range. The green rhombuses represent the mean of each box. ns: $p > 0.05$; *: $p \leq 0.05$; **: $p \leq 0.01$; ***: $p \leq 0.001$; ****: $p \leq 0.0001$. All the statistical tests in (**c**)−(**g**) are two-sided Mann-Whitney test, no adjustments were made for multiple comparisons.

eQTac method in another independent dataset[14] containing genotype, RNA-seq, and ATAC-seq data from 92 individuals (see "Methods" section). As shown in Fig. 5b, the area under the ROC (receiver operating characteristic) curve (AUC) was 0.81, supporting the robustness of our method. The whole pipeline of eQTac calculation was integrated into the package https://github.com/JFF1594032292/eQTac. Among the 164,378 PRE-gene pairs, 2047 pairs met the threshold of FDR < 0.05 (Supplementary Data 5, Supplementary Fig. 18).

Compared with eQTLs, eQTacs are more frequently found in the upstream and 5'UTR regions of genes (Fig. 5c). Using the chromatin

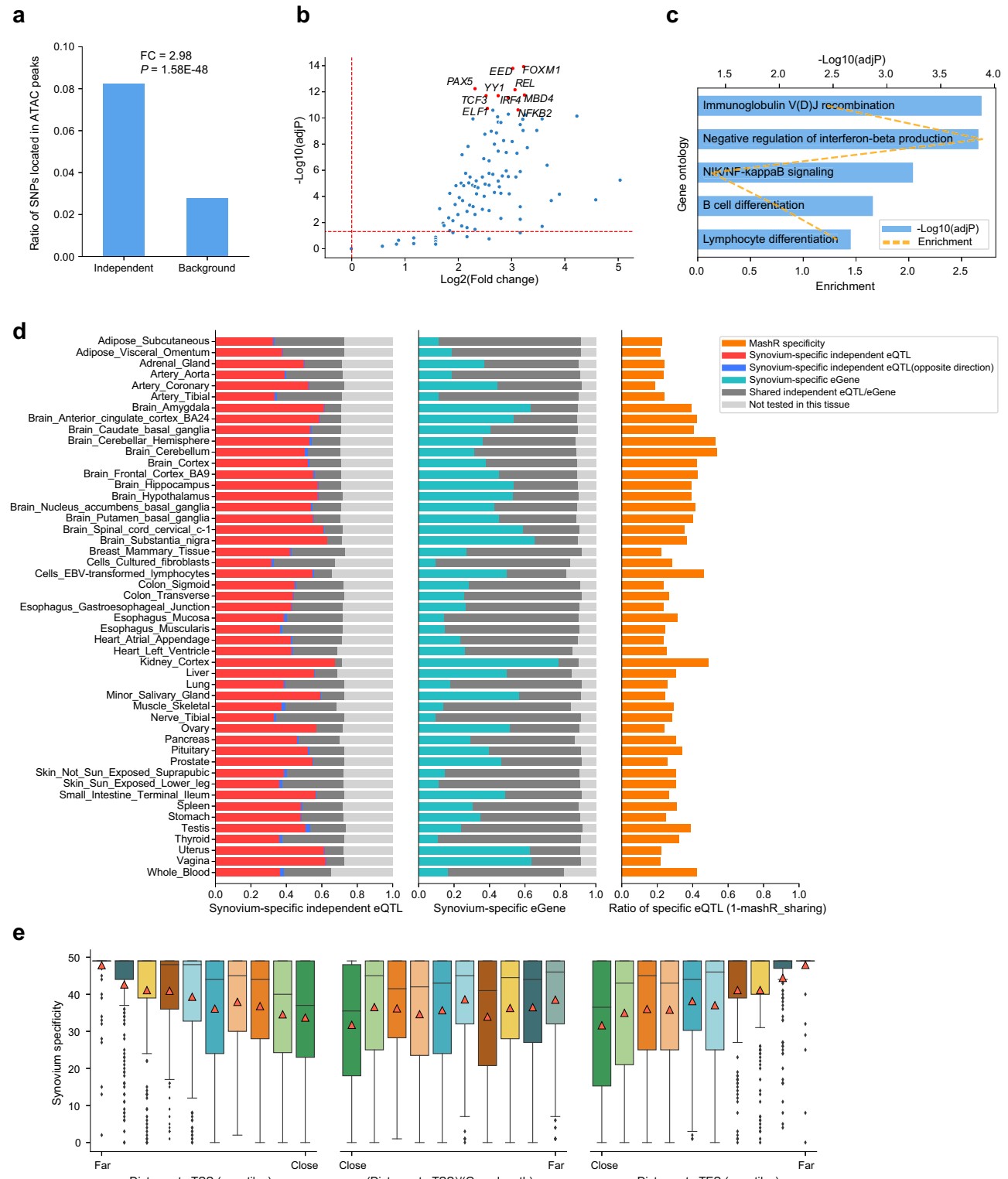

**Fig. 3 | Tissue specificity of synovium-independent eQTLs. a** Bar plot showed the independent SNPs and background SNPs locate in synovium ATAC peaks. Y-axis represents the ratio of SNPs locate in peaks. $P = 1.58 \times 10^{-48}$, enrichment FC = 2.98, two-sided Fisher exact test. **b** The enrichment of 120 TFs binding sites for in independent eQTLs. Top 10 enrichment TFs were labeled as red. Dash lines represented the adjusted $P = 0.05$ and log2(fold change) = 0, respectively (two-sided Fisher exact test). **c** The Gene Ontology (GO) enrichment for the top 10 enriched TFs. Two-sided hypergeometric tests were conducted, and adjustments were made for multiple comparisons. **d** Synovium-specific eQTLs in 49 GTEx tissues. The independent eQTLs that not significant (red bar), or significant but with opposite direction in the other tissue (blue bar) were defined as synovium-specific independent eQTL. **e** Synovium-specificity in different locations relative to their target gene. The synovium-specificity were defined as the count of not shared tissues for each synovium-independent eQTLs. The position of upstream SNPs was relative to target gene TSS (left panel), the position of downstream SNPs was relative to target gene TES (right panel), and the position of SNPs located on the target gene was the (distance to TSS) / (target gene length) (center panel). $N = 5380$ independent eQTLs. The boxplots represent 25th, 50th (median), and 75th percentiles, and whiskers extend to 1.5 times the interquartile range. Orange triangles represent mean values.

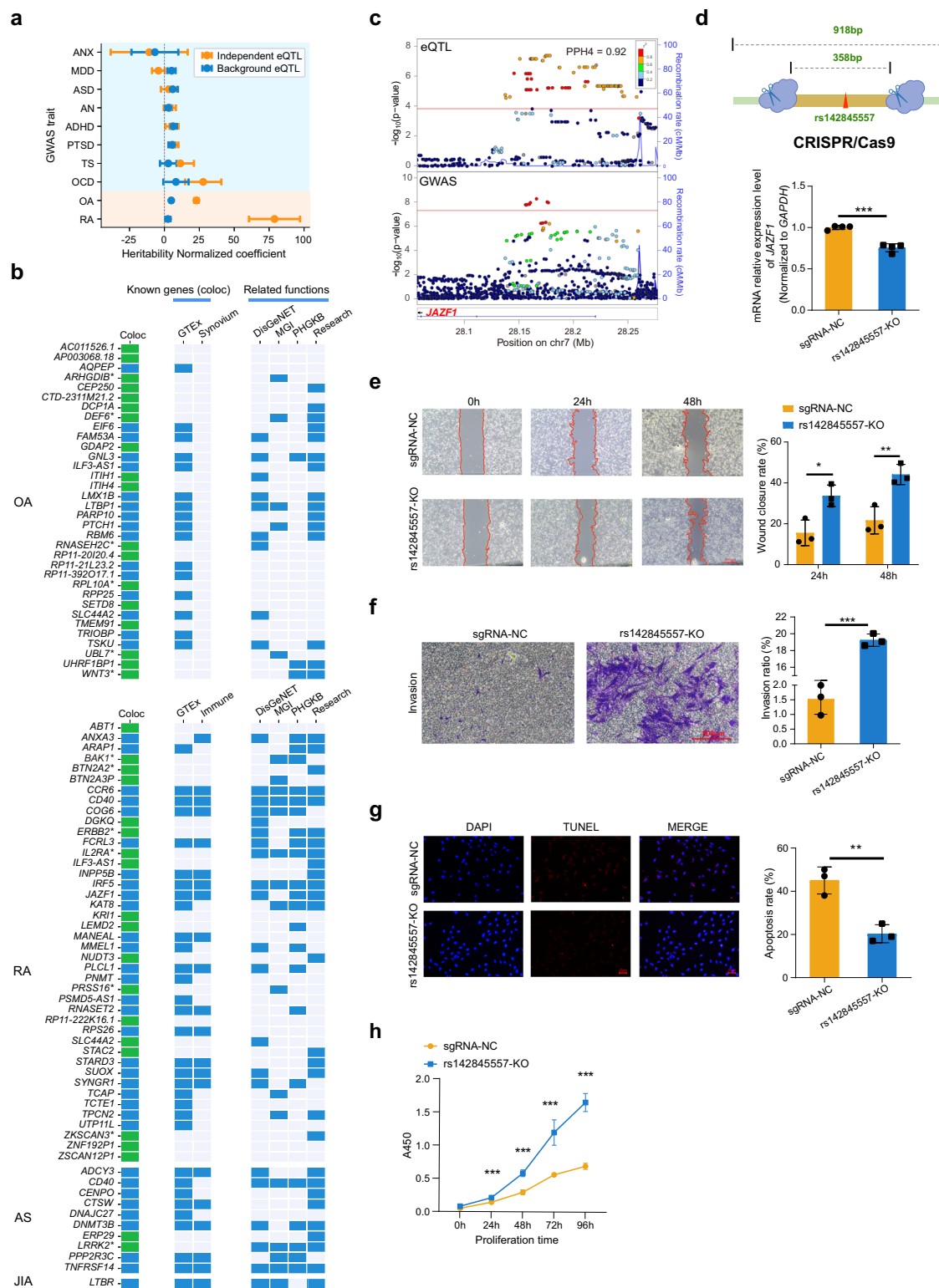

states annotation (HMM18) and synovium capture Hi-C (cHi-C) data (Supplementary Table 7), we observed that eQTacs are significantly enriched in promoter, enhancer and loop regions (Fig. 5d, Supplementary Fig. 19). Additionally, the eQTac-gene pairs overlapped significantly with enhancer-promoter pairs predicted by the ABC model[47] and EpiMap project[48] in all 131 and 31 tissues (Supplementary Table 7), compared to the non-significant PRE regions (Fig. 5e, paired-samples T-test, $P = 1.08 \times 10^{-127}$ and $P = 2.12 \times 10^{-14}$ respectively). Moreover, SNPs located in eQTacs were more likely to be causal SNPs of target genes

identified by dap-g (Fisher exact test, Fig. 5f), compared with non-significant PREs SNPs. These results demonstrated that eQTac could reflect the regulatory relationship between the functional open chromatin and target genes.

We estimated the heritability for the eQTac regions compared to non-significant PREs and found specifically enriched heritability for OA and RA GWAS loci (Fig. 5g), indicating that a certain extent of GWAS loci could be explained by eQTac. Over 30% of eQTac regions (477/1517) didn't contain significant eQTLs, suggesting that eQTac

**Fig. 4 | Colocalization of eQTLs and synovium-related GWAS loci. a** Enrichment of independent eQTL variants in synovium-related and mental disorder GWAS loci. OA and RA results were combined separately by meta-analysis. Coefficients from LD score regression were normalized by the per-SNP heritability (h2/total SNPs per GWAS), with points indicating the estimated coefficients and horizontal error bars indicating standard error (SEM). $N = 4794$ for independent eQTLs, $n = 14{,}308$ for background eQTLs. Background SNPs were generated from SNPsnap database with matched MAF, LD buddies, distance to nearest gene, and gene density. ANX anxiety disorders, MDD major depressive disorder, ASD autism spectrum disorder, AN anorexia nervosa, ADHD attention deficit hyperactivity disorder, PTSD post-traumatic stress disorder, TS Tourette syndrome, OCD obsessive-compulsive disorder. The utilized GWASs are described in Supplementary Data 2. **b** The 84 genes identified by colocalization analysis. Novel genes that didn't identified by previous eQTL colocalizations are shown in green. Novel genes that showed immune-related functions are labeled "*". DisGeNET, MGI, and PHGKB are the corresponding databases that showed related functions for colocalized genes. Research showed related functions from published articles. OA osteoarthritis, RA rheumatoid arthritis, AS ankylosing spondylitis, JIA juvenile idiopathic arthritis. **c** *JAZF1* eQTL and GWAS locus plots. Red lines represent the corresponding threshold of eQTL or GWAS association *P*-values. Points indicated the logistic (for GWAS) and linear regression (for eQTL) *P*-values, and adjustments were not made for multiple comparisons. **d–h** Effects of deletion of the region containing rs142845557 by CRISPR-cas9 on *JAZF1* mRNA expression levels ($P = 1.10 \times 10^{-4}$). **d** migration ($P = 1.86 \times 10^{-2}$ for 24 h, $P = 9.39 \times 10^{-3}$ for 48 h), **e** invasion rate ($P = 4.39 \times 10^{-6}$), **f** apoptosis ($P = 4.70 \times 10^{-3}$), **g** proliferation ($P = 1.28 \times 10^{-7}$, $P = 6.30 \times 10^{-7}$, $P = 1.03 \times 10^{-5}$, $P = 1.54 \times 10^{-8}$ for 24 h, 48 h, 72 h, 96 h, respectively), and **h** of MH7A cells. Bars denoted mean values and error bars denoted SD from one experiment performed in triplicate. *P*-values were determined with a two-tailed *t*-test. \*$P < 0.05$, \*\*$P < 0.01$, \*\*\*$P < 0.001$. Panel **d** was partly generated using Servier Medical Art, provided by Servier, licensed under a Creative Commons Attribution 3.0 Unported License.

could capture some regulatory effects beyond those identified by eQTLs. For example, the accessibility of an eQTac (chr6:33581434-33581934) was positively associated with expression of *BAK1* ($P = 4.60 \times 10^{-4}$, Fig. 5h), which is a RA colocalized gene (Fig. 4b) and have been reported to be involved in immune-related process. Interaction between this region and the *BAK1* promoter was also detected both in ABC and EpiMap prediction analysis (Fig. 5i). However, all the four SNPs located in this region were not significant eQTL (Supplementary Fig. 20).

Taken together, these findings suggest that some GWAS SNPs could regulate target gene expression through altering the accessibility of local chromatin. Our prediction of eQTac with only genotype and gene expression data could help identify regulatory chromatin accessibility on target genes in population level at a low-cost and convenient manner.

## Discussion

In this study, we provided a comprehensive molecular profile for synovium based on the genomic and transcriptomic features of human synovium in up to 245 OA patients. These data could facilitate future studies pinpointing causal disease variants and discovering the regulatory mechanisms underlying arthritic diseases and related immune diseases.

With much larger sample size than the previous study[11] (245 vs 77), our work provides an eQTL resource for understanding the role of synovium in arthritis. We identified 5381 independent eQTLs and 4765 eGenes, which is much more than the previous study[11]. We found that the eQTL variants are enriched in TF binding sites in open chromatin regions, and the top 10 TFs are enriched in immune-related gene ontology (GO) terms, such as immunoglobulin V(D)J recombination. Specifically, 7 of the 10 TFs (including *PAX5*[49,50], *TCF3*[51–53], *ELF1*[54,55], *REL*[56–58], *IRF4*[59], *YY1*[60], and *NFKB2*[61,62]) have been reported to be involved in the immune cell differentiation or immune response process. For example, *IRF4* could promote CD8$^+$ T cell exhaustion and limit the development of memory-like T cells during chronic infection[59]. Consistently, synovium indeed play important roles in immune-related process in the pathology of arthritis diseases. For example, RA is associated with the autoimmune process in synovium and the transformation into invading pannus[63], while OA pathogenesis implicates the release of mediators from synovium that lead to activation of different inflammatory pathways that damage cartilage[64–66].

We identified 84 colocalized genes by integrating our identified synovium eQTLs with GWAS summary data for multiple arthritis diseases. Functional experiments validated the effect of one eSNP rs142845557 on its target colocalized gene *JAZF1* expression and cellular phenotypes. *JAZF1* encodes a nuclear protein with three C2H2-type zinc fingers, and it was reported that *JAZF1* could limit chronic inflammation by reducing macrophage and CD4$^+$T cell populations,

and regulating the secretion of immune-related factors[67]. Notably, 38 colocalized genes are novel genes which have not been identified by previous studies using eQTL datasets from GTEx or other immune cells. In the PPI network constructed by the 84 colocalized genes, most of the novel genes formed interactions with well-known genes, suggesting that these genes might be closely related to influence the development of diseases together. For example, two hub genes, *ERBB2* and *LRRK2* are novel genes specifically identified by our synovium dataset. Although their roles in the pathogenesis of arthritis have not been reported before, previous studies have reported their involvement in immune-related process, such as cooperation with TGF-beta pathway[68–70], and immune response exacerbation[43]. In addition, another novel gene, *IL2RA* interacts with the well-known autoimmune disease genes *CCR6*, *IRF5*, and *CD40*, which are the hub genes in the network. *IL2RA* encodes the receptor for interleukin 2 and is involved in the regulation of immune tolerance by controlling regulatory T cells (TREGs) activity[71,72]. Our results provide evidence for future studies which aims to explore the potential mechanisms of these novel genes.

We developed the eQTac method, which could identify variants that affect gene expression by affecting chromatin accessibility without population-scale ATAC-seq data. The eQTac-gene pairs we obtained showed significantly higher proportion of overlap with the previous identified enhancer-genes or chromatin interactions. Validation analysis in another independent dataset supported the robustness of our method. Previous studies have shown that epigenomic state of a DNA regulatory element is specified primarily by its sequence[73]. Noncoding SNPs that disrupt open chromatin or enhancer function do so directly through modulation of local transcription factor-DNA interactions, leading to concomitant changes in chromatin state and gene expression[74]. Therefore, our eQTac can capture the variants that could affect gene expression by affecting chromatin state in a cost-effective manner, facilitating future studies which aim to investigate the regulatory mechanisms of variants contributing to disease development.

In summary, our study integrated multi-omics data from a large cohort of patients, and provided valuable insights into the regulation of gene expression and the diverse roles of the synovial tissue in diseases. We also developed a novel method, eQTac, for conveniently predicting regulatory elements associated with target genes using eQTL datasets, which has the potential to inspire further research. Our findings contribute to a deeper understanding of synovial tissue biology and provide a framework for investigating gene regulation in other contexts.

## Methods
### Ethics statement
The study was approved by the Ethics Committee of Xi'an Jiaotong University Honghui Hospital. All patients were provided written, informed consent before participating in the study.

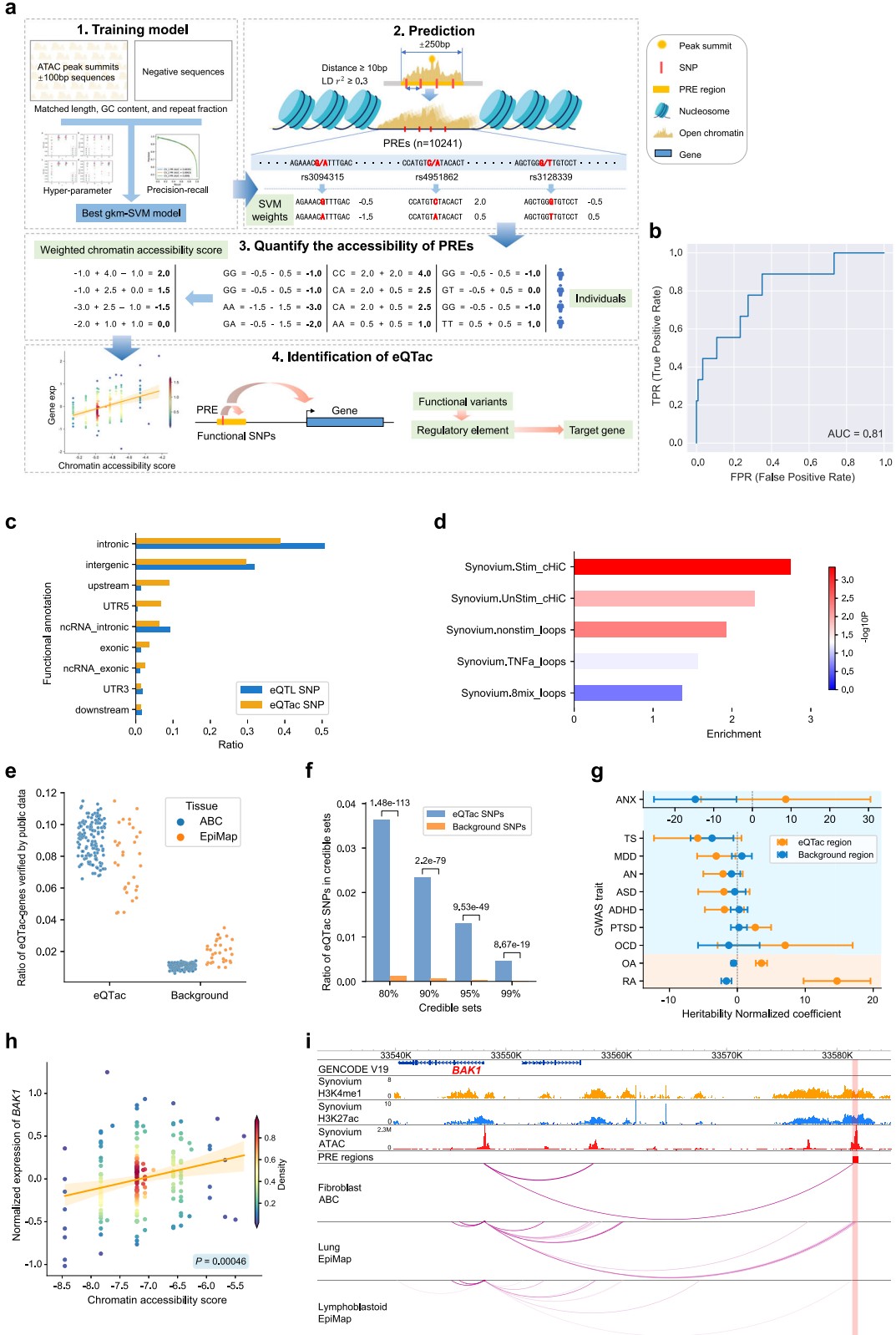

## Study participants and samples collection

We collected synovium samples and corresponding blood samples from 245 osteoarthritis patients undergoing knee joint replacement surgery (77 men, 168 women, age 46–84 years, mean 67 years), with no history of significant knee surgery, infection, or fracture, and no malignancy within the previous 5 years at the Honghui Hospital (Xi'an, China). All the patients were unrelated Chinese Han adults. Clinical information, including age, sex, and past medical history, was collected from the electronic medical records.

We obtained synovium samples from joint replacement and removed the nearby adipose tissue, then the samples freeze immediately with liquid nitrogen and stored under −80 °C. We also obtained the blood sample to extract DNA for genotyping for all patients.

**Fig. 5 | Identification of expression quantitative trait accessible chromatin (eQTac). a** The flowchart of eQTac: (1) An SVM model is trained with the positive sequences (ATAC-seq peaks) and matched negative sequences. (2) Potential regulatory elements (PRE) were selected and used trained SVM model to score the PRE variants. (3)The accessibility score for each PRE was calculated by weighted sum all variants scores. (4) eQTac (significant PRE-gene correlations) was identified through linear regression analysis for each PRE and gene pairs. **b** The ROC (receiver operating characteristic) curve (AUC) of eQTac method. **c** Comparison of variants annotations between eQTac SNPs and eQTL SNPs. **d** The enrichment of chromatin interaction in eQTac. Five datasets were capture Hi-C (cHi-C) or chromatin loops from synovium of OA patients (Supplementary Table 7). Two-sided Fisher exact test, no adjustments were made for multiple comparisons. **e** The overlap ratio between significant eQTac and enhancer-gene pairs which predicted in the ABC model and EpiMap database. Each point represents one tissue (Supplementary Table 7). Background pairs were non-significant eQTac-gene pairs. **f** The overlapped ratio between eQTac SNPs and fine-mapped SNPs in 80%, 90%, 95%, and 99%

credible sets, two-sided Fisher exact test. Significant eQTL SNPs were removed from eQTac SNPs. Background SNPs were non-significant eQTac SNPs and filtered by SNPsnap. **g** Enrichment of eQTac regions in synovium-related and mental disorder GWAS loci. Coefficients from LD score regression were normalized by the per-SNP heritability (h2/total SNPs per GWAS), with points indicating the estimated coefficients and horizontal error bars indicating standard error (SEM). Background regions were non-significant eQTac regions. OA and RA results were combined separately by meta-analysis. $N = 1517$ and 8351 for significant eQTac and background eQTac, respectively. Utilized GWASs are described in Supplementary Data 2. **h** Scatter plot showed the correlation between chromatin accessibility score of the PRE (chr6:33581434-33581934) and expression. Colors indicate the points density. Orange line represents the best-fitting linear regression, translucent bands around the regression line was 95% confidence interval for the estimated regression effect. Two-sided $t$-test for the regression effect. **i** Epigenetic annotation for *BAK1* eQTac result.

## DNA extraction

The genomic DNA was isolated from the blood using FlexGen Blood DNA Kit (KANGWEISHIJI Biotech, Jiangsu, China) according to the manufacturer's instructions.

## Genotyping and quality control

We used Illumina Infinium Asian Screening Array-24+ v1.0 for genotyping 659,184 variants (Engage to Life Energy Co. Ltd., China). Genotypes were called using GenomeStudio (Illumina) and transformed to plink format. The quality control was carried out referring to the GTEx project[6] and eQTLQC[75], which contained the following steps:

1. All variants were annotated to 1000 genome EAS data and removed variants which not inconsistent with 1000 genome.
2. Filtering out variants and individuals with a missing rate greater than 0.2, then filtered out variants and individuals with a missing rate greater than 0.02. Variants filtering should be performed before individual filtering. This was done in two steps to avoid too strict filtering.
3. Removed wrong sex individuals by "--check-sex" in plink 1.9.
4. Removed variants with MAF < 0.01 and Hardy-Weinberg equilibrium exact test $P$-value < 1e-6 by "--hwe 1e-6 midp" in plink 1.9.
5. Removed individuals with heterozygosity rate deviated more than 3 standard deviations from the heterozygosity rate mean. The heterozygosity rate was calculated by "--het" in plink 1.9, and this calculation was performed on variants in approximate linkage equilibrium ("--indep-pairwise 50 5 0.2" in plink 1.9).
6. For each pair of 'related' individuals with a pihat > 0.2, we removed the individual with the lowest call rate.

The final datasets contained 243 patients and 485,437 variants.

## Imputation

The imputation was conducted using Impute2 (v2.3.2_x86_64_static)[76,77] with the 1000 genomes haplotypes phase 3 data (https://mathgen.stats.ox.ac.uk/impute/1000GP_Phase3.html) as the reference panel. To increase the overall accuracy, we set -k 100 and -buffer 300, and other parameters were set to default.

We also conducted quality control for imputed variants as above (genotype missing rate, MAF, and Hardy-Weinberg equilibrium), expected the threshold of MAF was 0.05. We excluded 2 patients due to the high heterozygosity rate. The resulting dataset contained 243 patients and 4,499,337 variants.

## RNA-seq sequencing and quality control

We performed a gene expression analysis on synovium samples from 210 patients. The RNA was extracted by TRIzol extraction method. Total RNA was used as input material for the RNA sample preparations. Sequencing libraries were generated using NEBNext® UltraTM RNA

Library Prep Kit for Illumina® (NEB, USA) following the manufacturer's recommendations. The library preparations were sequenced on an Illumina Novaseq 6000 platform and 150 bp paired-end reads were generated (Novogene Co. Ltd. Beijing, China).

We used Fastp[78] (version 0.19.7) to perform basic statistics on the quality of the raw reads. The steps of data processing were as follows:

(1) Discard a paired reads if either one read contains adapter contamination.
(2) Discard a paired reads if more than 10% of bases are uncertain in either one read.
(3) Discard a paired reads if the proportion of low-quality (Phred quality < 5) bases is over 50% in either one read.

## Quantification of RNA levels and gene expression

We applied FastQC to check samples quality and excluded the samples with low Q20 and Q30. Clean RNA-seq reads were mapped to human reference genome hg19 (contains only autosomes, sex chromosomes and mitochondrial chromosomes) with STAR (v2.7.9a)[79] aligner based on the GENCODE v19 (July 2013 freeze) annotation, the parameters setting was same as GTEx v8 project pipeline[6]. Gene-level expression read counts and TPM values were generated by RNA-SeQC (v2.3.5)[80] with default parameters. The read mapping rates and base mismatch rates were calculated from the output of SAMtools (v1.9)[81] stats subcommand. The proportion of reads genomic origin were produced with QualiMap (v.2.2.2-dev)[82]. Samples should meet the following metrics as used for GTEx project: read mapping rate ≥ 0.2, base mismatch rate ≤ 0.01, intergenic mapping rate ≤ 0.3, rRNA mapping rate ≤ 0.3.

After removing samples that failed mapping quality control, we conducted expression outlier filtration. Briefly, the pairwise expression correlation coefficients were calculated using the log-transformed TPM values of all genes, assume the correlation coefficient between sample $i$ and sample $j$ is expressed as $r_{ij}$, we calculated

$$\bar{r}_i = \sum_j \frac{r_{ij}}{n} \tag{1}$$

The average correlation coefficient of sample $i$ with all others of the total n samples. Lower $\bar{r}_i$ represent a lower quality. Then we calculated

$$D_i = \frac{\bar{r}_i - \bar{\bar{r}}}{\text{median}(|\bar{r}_i - \bar{\bar{r}}|)} \tag{2}$$

to provide a sense of distance from the grand correlation mean $\bar{\bar{r}}$. Six samples with $D < -5.0$ were considered as outliers and removed. The filtered gene expression dataset included 204 synovium samples.

Finally, samples that passed both genotype data and RNA-seq data quality control were used for the following analysis. Gene expression values for all samples were normalized for eQTL analyses using the following procedure: (1) read counts were normalized between samples using TMM[83] implemented in edgeR;[84] (2) genes were selected based on expression thresholds of ≥ 0.1 TPM in ≥ 20% of samples and ≥ 6 reads (unnormalized) in ≥ 20% of samples; (3) expression values for each gene were inverse normal transformed across samples. Only autosomal genes were used in the following eQTL analysis.

## Covariates for eQTL analysis

To control for population effects on the discovery of QTLs, genotype principal components (PCs) were used as covariates in QTL mapping. The PCs were calculated by smartpca in EIGENSOFT[85] in 243 individuals, then we calculated the statistical significance of each principal component by twstats (Tracy–Widom statistics) and select the first two PCs as covariates ($P < 0.05$). To infer hidden factors associated with the cohort, sequencing batch, or other technical differences, we applied probabilistic estimation of expression residuals (PEER) for normalized expression data (run_PEER.R in GTEx v8 pipeline). According to the GTEx v8 method, 30 PEER factors were selected (150 ≤ samples count <250). Finally, sex and age were also used as covariates.

## Identification of cis-eQTLs

For eQTL analysis, we got 202 synovium samples with matched genotype and gene expression datasets. For each gene, we considered genetic variants within 1 Mb of the transcription start site (TSS) as cis-eQTL and followed a similar method with GTEx project[6]. All variants on autosome with MAF ≥ 0.05 across the 202 individuals were included, except the MHC region (chr6:28477797-33448354).

We used the GTEx modified version of FastQTL[86] (https://github.com/francois-a/fastqtl; gtex_v6p version) to calculate cis-eQTL, and the adaptive permutation mode was used with the setting --permute 1000 10000. Nominal $P$-values for each gene-variant pair were calculated based on linear regression, including all covariates. The gene-level $q$-values[87] were calculated based on the beta distribution-extrapolated empirical $P$-values from FastQTL. A false discovery rate (FDR) threshold of ≤0.05 was applied to identify genes with at least one significant cis-eQTL ("eGenes").

To identify all significant variant-gene pairs associated with cis-eGenes, the nominal $P$-value threshold was calculated as $F^{-1}(p_t)$, where $F^{-1}$ is the inverse cumulative distribution of the beta distribution and $p_t$ was the empirical $P$-value of the gene closest to the 0.05 FDR threshold. For each eGene, significant eQTLs were defined as variants with a nominal $P$-value below the nominal $P$-value threshold for that gene.

## Annotation of variants

The annotation of SNPs was performed by ANNOVAR[88] (version 2020 Jun 08), with annotation datasets wgEncodeGencodeBasicV19.

## Comparison of pLI over gene expression bins

This analysis was performed referring to eQTLGen[7] project. All genes were divided into 10 bins according to the average expression quantile. The pLI of each gene were downloaded from https://static-content.springer.com/esm/art%3A10.1038%2Fnature19057/MediaObjects/41586_2016_BFnature19057_MOESM241_ESM.zip[89].

## Identification of independent eQTL

We identified the conditionally independent eQTL signals using the forward stepwise regression followed by a backward selection step stepwise procedure described in GTEx v8[6], which was calculated by tensorQTL v1.0.6[90]. The primary eQTLs were defined as independent eQTLs with the highest ranking of each eGene, and the rest of the independent eQTLs were secondary independent signals.

## Estimation the variance explained by independent eQTLs in gene expression

We used GCTA[91] --make-grm to calculate the genetic relationship matrix for our samples, and then used --reml to calculate the variance explained by each independent eQTLs.

## Heritability estimation of eGenes

We compared the heritability of eGenes with single and multiple independent signals. We used GCTA[91] --reml to estimate the variance explained by the SNPs for each eGenes as the estimated heritability, which following the method of FUSION pipeline[92].

## ATAC-seq sequencing and peak calling

ATAC-seq libraries were constructed for synovium from following the original protocol[93]. In brief, two hundred thousand cells were lysed with cold lysis buffer (10 mM Tris-HCl, pH 7.4, 10 mM NaCl, 3 mM MgCl2, and 0.03% Tween20), and centrifuged at $500 \times g$ for 8 min at 4 °C. The supernatant was carefully removed, and the nuclei was resuspended with Tn5 transposase reaction mix (25 μl 2 × TD buffer, 2.5 μl Tn5 transposase, and 22.5 μl nuclease-free water) (Illumina) at 37 °C for 30 min. Immediately after the transposition reaction, DNA was purified using a Qiagen MinElute kit. Libraries were sequenced on an Illumina HiSeq X Ten sequencer. The ATAC-seq experiment and library sequencing were performed by Frasergen Bioinformatics Co., Ltd, Wuhan, China.

Adapters were trimmed from ATAC-seq reads sequences using custom Python scripts. Pair-end reads were aligned to hg19 using Bowtie2[94]. Duplicate reads and reads with MAPQ < 30 were discarded. After filtering, the qualified reads were subjected to MACS2[95] to call peaks for each sample with parameters (-q 0.05 --nomodel --shift -100 --extsize 200 --keep-dup all). In total, we identified 154,649 ATAC-seq peaks from synovium.

## Tissue specificity analysis

Synovium eQTLs were compared with 49 tissues from the GTEx v8 project (https://console.cloud.google.com/storage/browser/gtex-resources). The mashR (version 0.2.73)[20] method was used to assess sharing of significant signals among each tissue. Specifically, we randomly selected 1 million eQTL pairs from each tissue as the null signal. Then we fitted the model using the mash() function and used get_pairwise_sharing() function to assess sharing of significant signals among each pair of tissues.

## Epigenetic markers enrichment for independent signals

The epigenetic datasets were downloaded from the GEO database. We downloaded 15 histone peak data for 6 histone markers (H3K4me1, H3K4me3, H3K27ac, H3K36me3, H3K27me3, and H3K9me3) from GSE163548 and GSE112655[96], 3 histone markers (H3K27ac, H3K4me1, H3K4me3) from NBDC database ID hum0207.v1[97]. All epigenetic data were generated from knee OA patients' synovium tissues.

The enrichment analysis was performed by chi-square test, compared with background SNPs from SNPsnap[98] with matched MAF, LD buddies, distance to nearest gene, and gene density.

## Estimated heritability enrichment for arthritis and mental disorders

The heritability enrichment analysis was performed following the method of Kosoy et al.[13]. Briefly, we used the partitioned heritability analysis of LDSC[23] to calculate the heritability enrichment, and the estimated coefficients from LD score regression are normalized by the per-SNP heritability (h2/total SNPs per GWAS). To enable comparisons of the regression coefficients across traits with a wide range of heritabilities, we chose to normalize by the per-SNP heritability and named this adjusted metric the "normalized heritability coefficient". The normalized heritability coefficients of mental disorders were from

single GWAS summary data, and the normalized heritability coefficient of OA and RA were combined separately by meta-analysis (METASOFT v2.0.0)[99].

For enrichment analysis in independent eQTL, background SNPs were selected from SNPsnap database with matched MAF, LD buddies, distance to nearest gene, and gene density. For enrichment analysis in eQTac regions, background regions were non-significant eQTac regions.

## Colocalization analysis between synovium eQTLs and related traits GWAS

**Bayesian colocalization analysis.** We searched synovium-related traits in the NHGRI-EBI GWAS Catalog (version e105_r2022-03-08)[100], and downloaded 25 GWAS datasets that had full summary statistics available (Supplementary Data 2). To examine colocalization between eQTLs and GWAS associations, we analyzed all 25 genome-wide significant signals by using coloc v5.2.0[101].

Specifically, for each genome-wide significant signal ($P < 5 \times 10^{-8}$), we considered the region spanning 100 kb on either side of the index variants and merged the overlapped region. Correlations (LD) between SNPs were calculated in the UK Biobank or 1000 genomes EAS population, depending on the GWAS population. We also harmonized the allele orders between SNP summary statistics and reference population, to avoid conflict. The MHC region (chr6:28477797-33448354) was excluded from GWAS summary data.

Firstly, we used runsusie() function to distinguish multiple causal variants and obtained the posterior probability for each variant, the coverage of credible sets was set to 0.3 to capture moderate signals. Then for each GWAS signal that overlapped with any eQTL signals, we conducted the colocalization analysis. We considered a 60% posterior probability of GWAS and eQTL shared association in the region (PPH4 ≥ 0.6) to indicate evidence of colocalization.

**LD-based colocalization analysis.** For genes that couldn't well fine-mapped by coloc5, we used previously described methods by conducting LD and conditional analysis[102] to perform colocalization. We performed an initial colocalization analysis based on LD between a lead GWAS variant and a lead-independent eQTL variant.

To get the lead GWAS SNPs, we extracted all SNPs that met the threshold of genome-wide significance ($P = 5 \times 10^{-8}$) from both 25 GWAS summary datasets and GWAS catalog database. The GWAS catalog SNPs were searched as "arthritis" and "synovium", and downloaded all associations except the tendon rupture phenotype. Then manually selected the trait for "rheumatoid arthritis", "osteoarthritis", "juvenile idiopathic arthritis", "synovitis", and "ankylosing spondylitis". To reduce redundancy, we next pruned the GWAS SNPs by plink --clump with an LD threshold $r2 = 0.7$.

We then performed the conditional analysis in the eQTL data by providing genotypes for the lead GWAS variant to regression model as a covariate. We considered signals to be colocalized if (1) the pairwise LD was high between the GWAS variant and eQTL variants ($r2 \geq 0.7$ in both in eQTL population and GWAS population) and (2) after conditioning on the GWAS variant, the lead eQTL variant no longer met the eQTL mapping threshold of eGene.

## Comparison of colocalized genes in different eQTL datasets

For the colocalized genes, we compared our results with genes from previous representative studies to identify novel genes specifically identified by our synovium dataset. For OA, the GWAS data were collected from Boer et al.[24], Zengini et al.[26], and Tachmazidou et al.[27] ($n = 826,690$, 327,918, and 455,221, respectively). The colocalized genes were identified by using eQTL data from 48 GTEx tissues, and the synovium eQTL study[11] from 77 individuals, respectively. The RA GWASs data were collected from Ishigaki et al.[25,28], Ha et al.[32], and

Okada et al.[29] ($n = 212,453$, 276,020, 311,292, and 103,638 respectively). The colocalized genes were identified by using eQTL data from 48 GTEx tissues, DICE immune cells eQTLs, and BlurPrint immune cells eQTLs (monocyte, neutrophils, and T cells), respectively. The AS GWASs data were collected from Ellinghaus et al.[31] ($n = 42,939$). The colocalized genes were identified by using eQTL data from peripheral blood. The JIA GWASs data were collected from Hinks et al.[30] ($n = 15,872$). The colocalized genes were identified by using eQTL data from LCLs, T cells, and fibroblast. For AS and JIA, we also conducted LD-based colocalization analysis for the GWAS tag SNPs with 48 GTEx tissues and BlurPrint immune cells eQTLs (monocytes, neutrophils, and T cells) to get the colocalized genes. All used GWAS studies are listed in Supplementary Data 2. We annotated the related functions of colocalized genes in three databases: DisGeNET[103], MGI[104], PHGKB[105], and previously published articles.

## Protein–protein (PPI) analysis for colocalized genes

We conducted PPI analysis for the colocalized gene using the online version of STRING database v11.5 with an interaction threshold of 0.3. We used Cytoscape to visualize the obtained PPI network and identified hub genes with highest degree in the network.

## Cell culture

The rheumatoid fibroblast-like synoviocyte line MH7A were cultured in DMEM medium (HyClone, USA) supplemented with 10% fetal bovine serum (Biological Industries, Israel), 100 units/mL penicillin, and 0.1 mg/mL streptomycin at 37 °C incubator with 5% $CO_2$. The MH7A cell line was obtained from Shanghai Guan&Dao Biological Engineering Co., Ltd and was authenticated using short tandem repeat (STR) profiling by scientific service at Beijing Tsingke Biotech (Beijing, China).

## Fragment deletion by CRISPR-Cas9

Genotyping of rs142845557 was conducted by PCR in MH7A cells. A 918 bp sequence centered on rs142845557 was PCR-amplified from MH7A genomic DNA using primers in Supplementary Table 6. To efficiently eliminate the fragment containing rs142845557, CRISPR-associated RNA-guided endonuclease Cas9 cleavage technology (CRISPR-Cas9) was used[106]. In brief, we first designed a set of single-guided RNAs (sgRNAs) targeting upstream and downstream of the enhancer fragment by using the CRISPR design platform maintained by the CRISPick (https://portals.broadinstitute.org/gppx/crispick/public). One pair of sgRNA was designed for this SNP (Supplementary Table 6). Oligonucleotides containing these sgRNAs were cloned into lentiCRISPR v2 plasmid (Addgene#52961).

## DNA and RNA isolation and real-time qPCR (qRT-PCR)

DNA was isolated using the TIANGEN Genomic DNA Extraction Kit (catalog no. DP304; TIANGEN Biotech, Beijing, China). Total RNA was isolated from the MH7A cells using fast 200 (Fastagen, China) and was reverse-transcribed into cDNA by using the PrimeScript RT Reagent kit (TakaRa, Japan). The qRT-PCR reaction was performed using the QuantiTect SYBR Green PCR Kit (QIAGEN, USA). We used GAPDH as an endogenous control to normalize the differences in samples. Primers of qRT-PCR are shown in Supplementary Table 6.

## Scratch wound-healing assay

$1 \times 10^5$ cells/well MH7A cells undergoing different treatments were cultured in a 24-well tissue culture plate. The straight wound in the middle of the culture was subsequently created by a sterile pipette tip after cells reached 100% confluence. After being washed with phosphate-buffered saline (PBS) twice to smooth the edge of scratch and to remove the floating cells, the cells were cultured in DMEM medium supplemented with 1% fetal bovine serum at 37 °C with 5% $CO_2$.

## Transwell assay

Transwell assay was performed to test the invasion ability of MH7A cells. 8-μm-pore transwell chambers (Corning, USA) with 20 μL Matrigel precoated on the upper transwell chamber were put on a 24-well plate. The MH7A cells undergoing different treatments were transferred to the serum-free medium of the upper transwell chamber, 600 μL medium with 15% fetal bovine serum was added to the lower chamber as a chemoattractant. After 48 h, the upper chamber was washed with PBS multiple times. Cells in the upper layer that had not migrated were removed with cotton swabs gently. Moreover, the transwell chamber was fixed in 4% paraformaldehyde solution for 15 min and stained by 0.1% crystal violet for 20 min. The cells were counted by an inverted optical microscope (NOVEL, China) and photographed.

## Cell proliferation assay by using Cell-Counting Kit-8 (CCK-8)

The MH7A cells undergoing different treatments ($5 \times 10^3$ cells/well) were seeded in 96-well plates. And then, the CCK-8 solution was added to each well at 24, 48, 72, and 96 h, and incubated for another 3 h in the incubator. After the incubation, we used a microplate reader (MULTISKAN FC, Thermo Scientific, USA) to detect the optical density (OD) value at 450 nm. There were 5 replicates in each group.

## Cell apoptosis assay using one-step TUNEL apoptosis assay kit

The MH7A cells undergoing different treatments ($2 \times 10^4$ cells/well) were seeded in 96-well plates. TUNEL-reaction was performed by using the one-step TUNEL apoptosis assay kit (Beyotime, China) according to the manufacturer's instructions. Each well was fixed with paraformaldehyde for 30 min and permeabilized with 0.1% Triton® X-100 for 15 min. The wells were then washed with PBS and incubated with TUNEL test solution for 1 h at 37 °C. After washed by PBS twice, we used DAPI to counterstain cell nuclei for 10 min at room temperature. Randomly chosen fields were captured by using microscope (NOVEL, China).

## Quantify the accessibility of potential regulatory elements

Firstly, we trained a gkm-SVM[73] model to predict the chromatin accessibility for different alleles of variants located on the potential regulatory elements. The gkm-SVM produced a scoring function characterized by a set of weights quantifying the contribution of each possible 10-mer to a region's chromatin accessibility in synovium tissue. All 154,649 peaks of ATAC-seq were further trimmed, and 100-bp sequences of summits were used as the positive training set to maximize the open chromatin signals. We then generated a negative training set by randomly sampling from the genome of regions that matched the length, GC content, and repeat fraction of the positive training set (gkm-SVM v0.8.0). To remove false negative regions as much as possible, we excluded any regions with $P < 0.2$ from the sampling. We then trained a gkm-SVM model by LS-GKM[44] to accelerate computation, with default parameters in the gkm-SVM model (word length $l = 10$, informative columns $k = 6$, and truncated filter $d = 3$) and measured the classification performance using ROC curves.

Then we applied this model to predict the scores of sequences which 9 bp surrounding all SNPs by "gkmpredict" in the LS-GKM[44]. The chromatin accessibility score (CAS) for each PRE was defined as the weighted sum of genotype dosage of sequences around SNPs located on this PRE, with the predict the scores as weight:

$$CAS = \sum_{i=1}^{n} \sum_{j=1}^{m} (S_{ij} \times G_{ij}) \tag{3}$$

The $n$ represent the SNP counts in the PRE region, $m$ is the number of alleles for SNP $i$, $G$ is the genotype dosage for allele $j$, $S$ is the score of $j$ allele predicted by the SVM model.

The potential regulatory elements (PRE) were defined as the ±250 bp region of summits of each peak, which referred to the enhancer definition of Activity by Contact (ABC) model[47]. We excluded the PREs that contained any indel or insert mutations to keep the consistent length of sequences. We also removed PREs for those containing SNPs less than 10 bp apart, as these SNPs genotypes would disturb the prediction of each other. Considering that if a PRE only contained a single variant or multiple variants in stronger LD, the variation of this PRE across the population would be highly collinear with this variant (or haplotype), which was not expected in the following analysis. So, we only retained PREs containing more than one independent SNP (LD $R^2 < 0.3$).

## Validation of the eQTac method in another independent dataset

We used the dataset from dbGap phs000815[14] (https://www.ncbi.nlm.nih.gov/projects/gap/cgi-bin/study.cgi?study_id=phs000815.v2.p1) containing genotype, RNA-seq, and ATAC-seq data from 92 individuals to validate our eQTac method. For the genotype data, we performed imputation analysis and then SNPs with MAF < 0.1 were removed due to the small sample size. For the ATAC-seq data, we used macs2 to call the peaks and extract the 100 bp sequences around the summits of peaks as the positive training set. The gene expression data were processed using the same pipeline as our eQTL calculation. The actual open chromatin scores of PRE regions were defined as the inverse normalized transformed reads counts from ATAC-seq data. Significant eQTL genes were subjected to subsequent analysis. The correlation results using linear regression analysis for the open chromatin scores and target gene expression values were used to define the ground truth. That's, PRE-gene pairs with $P < 0.05$ were defined as true correlation. Then we used our eQTac method to identify the significant PRE-gene pairs.

## Identification of eQTac

The *cis*-eQTac was calculated for each gene calculated in *cis*-eQTL mapping, with the same *cis* region and covariates as eQTL mapping:

$$\text{expression} \sim \beta_1 \times \text{CAS} + \text{Covariates} \tag{4}$$

The CAS was the chromatin accessibility score for each PRE, with $\beta_1$ as the effect of this open chromatin region to the target gene. To identify the significant eQTac and controlled FDR (false discovery rates), we generated the null distribution of $P$-values for all PRE-gene pairs by randomly permuting the individual labels of gene expression 100 times. The FDR was determined as the proportion of permuted $P$-values over the proportion of non-permuted $P$-values under a specific *P-value* threshold. The threshold of $P$-values was set as $6.36 \times 10^{-4}$ under FDR = 0.05.

## Estimated causal SNPs for synovium eQTL

Dap-g[107] methods were applied to the *cis*-eQTL data to produce estimates of the causal SNPs. Dap-g is designed for multi-SNP genetic association analysis which employs a spike-and-slab prior model to select potential multiple independent *cis*-eQTLs in eQTL mapping. Briefly, the fine-mapping was conducted in the following steps:

(1) Calculated single-SNP Bayes factor for each SNP-gene pair.
(2) Calculated the priors probability for each SNP-gene pair by torus[108], which includes the distance of each SNP to gene TSS and the chromatin accessibility annotation as prior information.
(3) Conducted multi-SNP fine-mapping by dap-g, with -ld_control 0.5 and --no_size_limit.

The 80%, 90%, 95%, and 99% credible set for each *cis*-eQTL consists of variants that include the causal variant with 80%, 90%, 95%, and 99% probability, respectively.

## Reporting summary

Further information on research design is available in the Nature Portfolio Reporting Summary linked to this article.

## Data availability

All relevant data support the key findings of this study are available within the article and its Supplementary Information files. The RNA sequencing data and ATAC-seq data generated in this study have been deposited in the Genome Sequence Archive (GSA-Human)[109] in National Genomics Data Center[110], China National Center for Bioinformation / Beijing Institute of Genomics, Chinese Academy of Sciences under accession code HRA004624, which are available under restricted access as they contain identifying participant information. Access can be obtained by request via the GSA-Human database. Requests including a formal research proposal indicating the use of data and planned analyses will be processed within two weeks. The data is only allowed for academic use. The raw genotype data are protected and are not available as they contain identifying participant information and the key genetic information of individuals. The processed independent eQTLs, colocalization results, and significant eQTac data are available at Supplementary Data. Other public data used in the study are listed in Supplementary Tables and Supplementary Data.

## Code availability

Scripts to eQTac method are available at https://github.com/JFF1594032292/eQTac and https://zenodo.org/records/10254586.

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

## Acknowledgements

This work was supported by the National Natural Science Foundation of China (32170616 (T.L.Y.), 32370653 (Y.G.), 82170896 (Y.G.), and 82372458 (S.S.D.)); Science Fund for Distinguished Young Scholars of Shaanxi Province (2021JC-02 (T.L.Y.)); Innovation Capability Support Program of Shaanxi Province (2022TD-44 (T.L.Y.)); Health Commission General Cultivation Program of Xi'an City (2021ms10 (Z.Y.)); Key Research and Development Project of Shaanxi Province (2023-YBSF-180 (Y.G.), 2022GXLH-01-22 (T.L.Y.)), and the Fundamental Research Funds for the Central Universities. This study is also supported by the High-Performance Computing Platform and Instrument Analysis Center of Xi'an Jiaotong University. We thank the participants and the organizers of the Genotype-Tissue Expression (GTEx) project. Parts of the Fig. 1a and Fig. 4d were drawn by using pictures from Servier Medical Art and changes were made to the pictures. Servier Medical Art by Servier is licensed under a Creative Commons Attribution 3.0 Unported License (https://creativecommons.org/licenses/by/3.0/).

## Author contributions

Conceptualization: T.L.Y. and Y.G.; data analysis: F.J. and W.T.; resources: S.Y.H.; methodology: F.J., W.T., and S.S.D.; visualization and software: F.J. and H.W.; data curation: F.J., W.T., X.F.C., N.Y., D.J.Z., H.M.S., J.G., H.W.G., and C.Y.H.; cell biology experiments: N.N.W., N.Y., D.J.Z., C.W., and D.L.Z.; writing - original draft preparation: F.J.; writing - review and editing: S.S.D., Y.G and T.L.Y.; project administration and supervision: T.L.Y. and Z.Y.

## Competing interests

The authors declare no competing interests.
