## [Peer Review File · Nature Communications]

A landscape of gene expression regulation for synovium in arthritisREVIEWER COMMENTS

Reviewer #1 (Remarks to the Author):

The authors studied the genetic and molecular features of synovium, a tissue that covers the joints and is affected by arthritis. They analyzed the data from 245 patients with different types of arthritis and found how genetic variants influence gene expression and chromatin accessibility in synovium. They also identified genes that are linked to arthritis risk and severity. Their study provides a valuable resource for understanding the role of synovium in arthritis. I have several comments on this study below:

1) The authors stated that the effects of lead SNPs were positively correlated with the coefficients of variation (CVs) in gene expression. Lowly expressed genes were less accurately quantified in RNA-seq and therefore had larger CVs. It is suggested that the authors classify genes into different groups according to their expression levels, and then examine the effect size of lead SNPs in these groups. This may help to rule out confounding effects of gene expression quantification on the effect size of lead SNPs.

2) “Significant enrichments were also observed in active histone markers (H3K27ac, H3K4me1, H3K4me3, H3K36me3), and a concomitant depletion in repressed marker (H3K9me3), using public histone modification data in synovium (Figure S3, Table S2).” eQTL associates genetic variants with the variation of gene expression levels. It is surprising that eQTLs were enriched in active histone markers, while were depleted in repressed markers. How to interpret these results?

3) In the results, 41.6% (2240/5381) independent eQTLs were not 162 significant in any GTEx tissues. To estimate the proportion of synovium eQTLs not 163 shared with other GTEx tissues, we used p-value enrichment analysis (π_0)[29] (see 164 methods). The estimated mean proportion of synovium eQTLs not shared with other 165 tissues was $\pi_0 = 0.37$ (range, 0.2 - 0.58), indicating the presence of a certain component 166 of tissue-specific genetic regulation that can be specifically detected in synovium. The π_0 ranged from 0.2 - 0.58, which contained “41.6%”. Does this result suggest tissue-specific genetic regulation?

4) Figure 4b showed pathways such as B cells, and T cells enriched for genes colocalized with RA, AS, and JIA. However, according to Table S6, these are not enriched pathways rather, gene ontology (GO).

5) In the results, the author used the prediction model to calculate chromatin accessibility score for these 10,241 PREs for each individual. While in the methods, the authors stated that we got 4761 PREs with quantified chromatin accessibility scores for 202 individuals. Please explain the discrepancies.

6) A more detailed legend is needed in Figure 5a. Has the model for determining PRE from open chromatin sequences and calculating chromatin accessibility scores been tested and validated before?

Reviewer #2 (Remarks to the Author):

The authors collected the synovium from 245 arthritic patients, performed genotyping and RNA sequencing. Combining genetics and transcriptomics data together, 87 arthritis related genes were identified, a new analysis method was developed and 1517 regions with potential regulatory function of chromatin accessibility had been discovered. This work enriched the human synovium sequencing dataset and found more arthritis related genes. In addition, this study brings an important contribution to comprehensive understanding the landscape of genetic regulation on gene expression in arthritis. Beside these merits, some points still need to be clarified to make this work more relevant to previous genetics and transcriptomics data from other arthritic patient's cohorts.

1. Whether there are any common SNPs and eQTL genes that had been found in this study and another work. (A molecular quantitative trait locus map for osteoarthritis. *Nat Commun.* 2021, 12(1):1309.). The author should clarify the common SNPs and eQTL genes in these two studies, if there were no common SNPs and eQTL genes, the potential reasons should also be clarified.
2. In this paper, the author developed eQTac method to detect the open chromatin region, whether this method could be used for analyzing the dataset from a different arthritic patient cohort.
3. PPI analyze should be performed to test the connections among those 87 genes.

Reviewer #3 (Remarks to the Author):

The authors provide a very nice resource of expression and eQTL in a relatively sizable understudied tissue. This will be of great utility to fields where synovium tissue is involved in disease. They have highlighted eQTLs in relevant genes and TF, and ones that are implicated in disease - particularly RA and OA, adding to previous literature on expression in this cell type. They have also developed an analysis method to relate ATAC-seq activity with gene expression

By incorporating publicly available data, the authors support concepts of gene expression regulation, and GWAS correlation. It would be nice to take some of this initial bioinformatic correlations and confirm some of their hypothesis with functional work in a similar tissue setting. For example confirming SNPs change the binding of TF or GWAS associated SNPs in open chromatin can change expression and cellular phenotype, and so move the field forward in terms of the understanding disease mechanism in a loci

Reviewer #4 (Remarks to the Author):

Jiang et al used RNA-seq and ATAC-seq to characterize the landscape of genetic regulation on gene expression in 245 OA patients. They identified 4765 independent primary and 616 secondary cis-eQTLs. Integration of GWAS and eQTLs identified 87 arthritis-related gene.

My main comment is that authors should be more explicit about how this is an advance on previous work, and what is the novelty. The authors have largely focused on the number of detected eQTLs and technical details, with little insights into what novelty is provided by their study. For example, although authors profile a synovium from OA patients, there is a high number of colocalizations in other joint diseases of autoimmune nature which are etiologically fundamentally different from OA (RA, JIA and AS). This is interesting, and perhaps not surprising, but there is no detailed analysis of how many of these colocs are new or have been seen by profiling individual immune cells and what is specific for their synovium dataset. The developed method is interesting, but needs an independent validation. The biology interpretations are profoundly lacking.

Other concerns:

1. I am a bit confused by the fact that in Figure 1 authors categorize continuous variables. In my opinion it either should be continuous or if distances and effect sizes are converted to categories those categories should be explained. Distance (near and far) and effect size (low and high) are extremely ambiguous. In 1C and 1D would it be better to perform a simple linear regression of two continuous variables? What is the meaning of comparing the two neighbouring violon plots?
2. I am failing to grasp the significance of Figure 3 and ATAC-seq analysis. It is well accepted that eQTLs need to be in active/open chromatin in order to express their effect on gene expression. What are the conclusions here and how this differs to any other eQTLs?
3. The sharing between synovium eQTLs and GTEX tissues is a bit unexpected. What would be the explanation for having more shared eQTLs with nerves and thyroid than with the kidney or brain? Have you tried more stringent methods (for example mashR PMID: 30478440)? Since you see many immune signatures why didn't you compare them with many datasets coming from immune cells (e.g. DICE, BLUEPRINT, OneK1k ...)?

4. The most novel part is the development of eQTac. Although it is very intriguing it needs an independent validation to make sure that these signals are real. There are several datasets that have eQTLs and ATAC-seq QTLs (e.g. PMID: 35591976; PMID: 29988122). In my opinion, authors should validate their method by comparing the data to the ground truth eQTac signals.

Response to reviewers' comments:

Reviewer #1:

The authors studied the genetic and molecular features of synovium, a tissue that covers the joints and is affected by arthritis. They analyzed the data from 245 patients with different types of arthritis and found how genetic variants influence gene expression and chromatin accessibility in synovium. They also identified genes that are linked to arthritis risk and severity. Their study provides a valuable resource for understanding the role of synovium in arthritis. I have several comments on this study below:

1. **Comment:** The authors stated that the effects of lead SNPs were positively correlated with the coefficients of variation (CVs) in gene expression. Lowly expressed genes were less accurately quantified in RNA-seq and therefore had larger CVs. It is suggested that the authors classify genes into different groups according to their expression levels, and then examine the effect size of lead SNPs in these groups. This may help to rule out confounding effects of gene expression quantification on the effect size of lead SNPs.

Response:

Thank you very much for your constructive comment! As you suggested, we classified the genes into three groups (low/medium/high) according to their expression levels quantiles and examined the correlations between the effect size of lead SNPs and CVs in gene expression. As shown in the following Figure, the results showed similar positive correlations between the effect size of lead SNPs and CVs in gene expression in the low and medium gene expression groups, but the correlation was not significant in the high expression group due to the smaller CVs in highly expressed genes. Therefore, to rule out the potential confounding effects of gene expression levels, we used gene expression as a covariate and repeat the analysis. As shown in updated Figure 1d, the overall trend remained similar to our previous results. We have updated the results accordingly (Line 109-111).

Figure. The Pearson correlation between effect size and CVs in different gene expression levels (low/medium/high). Lead SNP effect sizes were split into 3 intervals according to the quantiles, larger quantile number indicates larger effect size values. The fitting line to the median values is shown in blue.

2. **Comment:** “Significant enrichments were also observed in active histone markers (H3K27ac, H3K4me1, H3K4me3, H3K36me3), and a concomitant depletion in repressed marker (H3K9me3), using public histone modification data in synovium (Figure S3, Table S2).” eQTL associates genetic variants with the variation of gene expression levels. It is surprising that eQTLs were enriched in active histone markers, while were depleted in repressed markers. How to interpret these results?

Response:

Thank you for your comment. We are sorry for the ambiguous word “repressed”. In fact, H3K9me3 is a defining modification of heterochromatin[1], which is a stable and silenced chromatin state. To be clear, we have replaced “repressed marker” with “heterochromatin” in the revised manuscript. We have also compared our results with previous eQTL studies in other tissues. Similarly, eQTL SNPs are enriched in the regions of transcriptionally active chromatin (e.g., H3K4me3, H3K27ac) and depleted in the heterochromatin (e.g., H3K9me3)[2, 3]. These results suggest that most cis-eQTL effects might be driven by genetic perturbations in regulatory elements in active

chromatin regions. We have added this statement in the results section (Line 159-165).

Reference

1. Padeken, J., S.P. Methot, and S.M. Gasser, *Establishment of H3K9-methylated heterochromatin and its functions in tissue differentiation and maintenance*. Nat Rev Mol Cell Biol, 2022. **23**(9): p. 623-640.
2. Caliskan, M., et al., *Genetic and Epigenetic Fine Mapping of Complex Trait Associated Loci in the Human Liver*. American Journal of Human Genetics, 2019. **105**(1): p. 89-107.
3. Eales, J.M., et al., *Uncovering genetic mechanisms of hypertension through multi-omic analysis of the kidney*. Nat Genet, 2021. **53**(5): p. 630-637.

3. **Comment:** In the results, 41.6% (2240/5381) independent eQTLs were not significant in any GTEx tissues. To estimate the proportion of synovium eQTLs not shared with other GTEx tissues, we used p-value enrichment analysis (pi)[29] (see methods). The estimated mean proportion of synovium eQTLs not shared with other tissues was $\pi_0 = 0.37$ (range, 0.2 - 0.58), indicating the presence of a certain component of tissue-specific genetic regulation that can be specifically detected in synovium. The π_0 ranged from 0.2 - 0.58, which contained “41.6%”. Does this result suggest tissue-specific genetic regulation?

Response:

Thank you for the comment. Yes, our results suggest the tissue-specific genetic regulation in synovium. There were 41.6% of the independent eQTLs showed significant signals (significant SNP-gene pairs, FDR < 0.05) in synovium, but not in any GTEx tissues. We further used the π_0 method to estimate the proportion of eQTLs specifically detected in synovium. This method takes all SNP-gene pairs into consideration when estimates the proportion of true null P -values in other tissues[4]. Therefore, the proportions obtained from these two methods were not exactly the same.

In the revised manuscript, in ref to the suggestion from reviewer #4, we used mashR[5] to estimate the proportion of eQTLs specifically detected in synovium. Instead of

using P values directly, this method used the SNP effect (beta and se) in the calculation model, improving effect estimates and allowing for more quantitative assessments of effect-size heterogeneity. The results showed that an average of 32% (range 19%-53%) synovium eQTLs were tissue-specific compared to the GTEx tissues (Figure 3d), supporting that our analysis could detect tissue-specific genetic regulation in synovium. We have revised the methods and results accordingly (Line 175-187, 590-596).

Reference

4. John D. Storey, R.T., *Statistical significance for genomewide studies*. PNAS, 2003.
5. Urbut, S.M., et al., *Flexible statistical methods for estimating and testing effects in genomic studies with multiple conditions*. Nat Genet, 2019. **51**(1): p. 187-195.

4. **Comment:** Figure 4b showed pathways such as B cells, and T cells enriched for genes colocalized with RA, AS, and JIA. However, according to Table S6, these are not enriched pathways rather, gene ontology (GO).

Response:

Thank you for the comment. We apologize that we misspelled “GO” as “pathway”. We have corrected this error in the revised manuscript (Line 229, 231, Supplementary Table S9).

5. **Comment:** In the results, the author used the prediction model to calculate chromatin accessibility score for these 10,241 PREs for each individual. While in the methods, the authors stated that we got 4761 PREs with quantified chromatin accessibility scores for 202 individuals. Please explain the discrepancies.

Response:

Thank you for pointing this out. We apologize for the inaccurate statement in the method section. In fact, we obtained 10,241 PREs with quantified chromatin accessibility scores for 202 individuals. We have removed the statement in the method section and this information is now shown in line 283.

6. **Comment:** A more detailed legend is needed in Figure 5a. Has the model for determining PRE from open chromatin sequences and calculating chromatin accessibility scores been tested and validated before?

Response:

Thank you for the comment.

We have added detailed legend for Figure 5a as follows:

(a) The flowchart of eQTac: 1) A SVM model is trained with the sequences in ATAC-seq peaks (positive) and matched negative regions. 2) Potential regulatory elements (PRE) were selected and the trained SVM model was used to score the variants located in PRE. 3) For each subject, the accessibility score for each PRE was calculated by weighted sum all variants located in the PRE. 4) eQTac (significant PRE-gene correlations) was identified through linear regression analysis for each PRE and genes with the distance of less than 1 Mb.

In the model, PREs was defined as the ± 250 bp region surrounding each peak summit from the ATAC-seq data. Then the SVM model was used to score the variants located in PRE and the accessibility score for each PRE was calculated by weighted sum all variants located in the PRE. The model for determining PRE from open chromatin sequences and calculating chromatin accessibility scores was validated with the three-fold cross validation process. As shown in Supplementary Fig.17, the best model achieved an area under the receiver operating characteristic curve (AUC) of 0.92. In the revised manuscript, we also tested the model in another independent synovium dataset[6], which contains ATAC-seq data from 11 subjects. As shown in Supplementary Fig. 17, the AUC ranged from 0.82 to 0.85, supporting the robustness of the model. We have added the corresponding method and results in the main text (Line 274-277).

Reference

6. Ai, R., et al., *Comprehensive epigenetic landscape of rheumatoid arthritis fibroblast-like synoviocytes*. Nat Commun, 2018. **9**(1): p. 1921.

Reviewer #2:

The authors collected the synovium from 245 arthritic patients, performed genotyping and RNA sequencing. Combining genetics and transcriptomics data together, 87 arthritis related genes were identified, a new analysis method was developed and 1517 regions with potential regulatory function of chromatin accessibility had been discovered. This work enriched the human synovium sequencing dataset and found more arthritis related genes. In addition, this study brings an important contribution to comprehensive understanding the landscape of genetic regulation on gene expression in arthritis. Beside these merits, some points still need to be clarified to make this work more relevant to previous genetics and transcriptomics data from other arthritic patient's cohorts.

1. **Comment:** Whether there are any common SNPs and eQTL genes that had been found in this study and another work. (A molecular quantitative trait locus map for osteoarthritis. Nat Commun. 2021, 12(1):1309.). The author should clarify the common SNPs and eQTL genes in these two studies, if there were no common SNPs and eQTL genes, the potential reasons should also be clarified.

Response:

Thank you for the comment! Following your suggestion, we have provided a Venn plot (Supplementary Fig. 1) to show the overlap between our results and the previous study[1] (Nat Commun. 2021, 12(1):1309). There are 27,013 common eSNPs and 462 common eGenes, respectively. We have added the results in the main text (Line 93-96).

Reference

1. Steinberg, J., et al., *A molecular quantitative trait locus map for osteoarthritis*. Nat Commun, 2021. **12**(1): p. 1309.

2. **Comment:** In this paper, the author developed eQTac method to detect the open chromatin region, whether this method could be used for analyzing the dataset from a different arthritic patient cohort.

Response:

Thank you for the comment! Yes, we developed the eQTac method to identify eQTLs that affect gene expression by affecting chromatin accessibility in the absence of population scale ATAC-seq data, therefore, this method could be used to other eQTL datasets. We collected another independent synovium dataset[2] containing genotype, RNA-seq, and ATAC-seq data from 92 individuals, to test our method. As shown in Figure 5b, the area under the ROC (receiver operating characteristic) curve (AUC) is 0.81, supporting the robustness of our method. We have added the corresponding method and results in the main text (Line 288-292, 795-808).

Reference

2. Ai, R., et al., *Comprehensive epigenetic landscape of rheumatoid arthritis fibroblast-like synoviocytes*. Nat Commun, 2018. **9**(1): p. 1921.

3. **Comment:** PPI analyze should be performed to test the connections among those 87 genes.

Response:

Thank you for the comment! As you suggested, we performed PPI analysis for the 84 genes using the STRING database[3]. As shown in Supplementary Fig. 10, the hub genes with the highest degree in the network (*CCR6*, *CD40*, *IRF5*, *ERBB2*, and *LRRK2*) (Supplementary Fig. 10) mainly participate in the immune-related process. *CCR6*, *CD40*, and *IRF5*, are well-known autoimmune disease genes associated with RA or AS[4-7]. Specifically, *ERBB2* and *LRRK2* are firstly identified by our synovium dataset as colocalized genes for RA and AS, respectively. *ERBB2* is a known oncogene with significant role in mediating tumor immune response[8, 9]. *LRRK2* is highly expressed in immune cells[10-12]. Mutant *LRRK2* could exacerbate immune response and neurodegeneration in a chronic model of experimental colitis[13]. Moreover, most of the novel colocalized genes identified by our synovium dataset formed interactions with known genes, suggesting that these genes might be closely related to influence the development of diseases together. Future functional

studies are encouraged to explore their potential mechanisms. We have added these descriptions in the results and discussion section (Line 210-226, 361-373).

Reference

3. Szklarczyk, D., et al., *The STRING database in 2023: protein-protein association networks and functional enrichment analyses for any sequenced genome of interest*. Nucleic Acids Res, 2023. **51**(D1): p. D638-D646.
4. Tawaraishi, T., et al., *Identification of a novel series of potent and selective CCR6 inhibitors as biological probes*. Bioorg Med Chem Lett, 2018. **28**(18): p. 3067-3072.
5. Grewal, I.S. and R.A. Flavell, *A central role of CD40 ligand in the regulation of CD4+ T-cell responses*. Immunol Today, 1996. **17**(9): p. 410-4.
6. Marzaioli, V., et al., *CD209/CD14(+) Dendritic Cells Characterization in Rheumatoid and Psoriatic Arthritis Patients: Activation, Synovial Infiltration, and Therapeutic Targeting*. Front Immunol, 2021. **12**: p. 722349.
7. Manni, M., et al., *Regulation of age-associated B cells by IRF5 in systemic autoimmunity*. Nat Immunol, 2018. **19**(4): p. 407-419.
8. Mei, J., et al., *Clinical and molecular immune characterization of ERBB2 in glioma*. Int Immunopharmacol, 2021. **94**: p. 107499.
9. Wu, S., et al., *HER2 recruits AKT1 to disrupt STING signalling and suppress antiviral defence and antitumour immunity*. Nat Cell Biol, 2019. **21**(8): p. 1027-1040.
10. Oun, A., et al., *LRRK2 protects immune cells against erastin-induced ferroptosis*. Neurobiol Dis, 2022. **175**: p. 105917.
11. Wallings, R.L. and M.G. Tansey, *LRRK2 regulation of immune-pathways and inflammatory disease*. Biochem Soc Trans, 2019. **47**(6): p. 1581-1595.
12. Tian, Y., et al., *LRRK2 plays essential roles in maintaining lung homeostasis and preventing the development of pulmonary fibrosis*. Proc Natl Acad Sci U S A, 2021. **118**(35).
13. Cabezudo, D., et al., *Mutant LRRK2 exacerbates immune response and neurodegeneration in a chronic model of experimental colitis*. Acta Neuropathol, 2023. **146**(2): p. 245-261.

Reviewer #3:

The authors provide a very nice resource of expression and eqtl in a relatively sizable understudied tissue. This will be of great utility to fields where synovium tissue is involved in disease. They have highlighted eQTLs in relevant genes and TF, and ones that are implicated in disease - particularly RA and OA, adding to previous literature on expression in this cell type. They have also development an analysis method to relate ATAC-seq activity with gene expression.

Comments:

By incorporating publicly available data, the authors support concepts of gene expression regulation, and GWAS correlation. It would be nice to take some of this initial bioinformatic correlations and confirm some of their hypothesis with functional work in a similar tissue setting. For example, confirming SNPs change the binding of TF or GWAS associated SNPs in open chromatin can change expression and cellular phenotype, and so move the field forward in terms of the understanding disease mechanism in a locus.

Response:

Thank you for your constructive comment!

We selected a RA associated SNP rs142845557 to validate whether it can influence target gene (*JAZF1*) expression and cellular phenotype. eQTL analysis showed that the allele A of rs142845557 was significantly associated with increased expression of *JAZF1* ($P = 7.3 \times 10^{-8}$, $\beta = 0.28$, Supplementary Fig. 13) with high probability of colocalization (PP.H4 = 0.92, Figure 4c). Epigenomic annotation analysis showed that rs142845557 is located in histone markers of active enhancer (H3K27ac and H3k4me1, Supplementary Fig. 14). The rs142845557 was homozygous AA associated with increased *JAZF1* expression in MH7A cells, hence we deleted a 358-bp genomic region containing rs142845557-AA using CRISPR/Cas9 in MH7A cells (Figure 4d). The deletion efficiency was confirmed by gel electrophoresis experiments (Supplementary Fig. 15b). As shown in Figure 4d, significantly decreased *JAZF1* expression ($P < 0.01$) was detected in rs142845557-AA deleted cells (KO) compared with the normal control cells, indicating the regulation role of this SNP on

JAZF1 expression.

We further conducted a series of functional experiments to examine the cellular phenotypes in rs142845557-KO cells, including migration, invasion, proliferation, and apoptosis. Compared with the control cells, wound-healing and transwell assays showed that the migration and invasion abilities of MH7A were significantly increased in the KO cells ($P < 0.05$) (Figure 4e and 4f). CCK-8 assay showed significantly enhanced cell proliferation ability in the KO cells (Figure 4g). TUNEL apoptosis experiment revealed significantly reduced apoptosis in the KO cells (Figure 4h). Taken together, our results reveal the regulatory effect of the SNP rs142845557 on target gene *JAZF1* expression and cellular phenotypes, highlighting the importance of this GWAS SNP involved in the pathogenesis of RA. We have added the corresponding results and methods in the revised manuscript (Line 238-261, 698-760). Thanks.

Reviewer #4

Jiang et al used RNA-seq and ATAC-seq to characterize the landscape of genetic regulation on gene expression in 245 OA patients. They identified 4765 independent primary and 616 secondary cis-eQTLs. Integration of GWAS and eQTLs identified 87 arthritis-related gene.

My main comment is that authors should be more explicit about how this is an advance on previous work, and what is the novelty. The authors have largely focused on the number of detected eQTLs and technical details, with little insights into what novelty is provided by their study. For example, although authors profile a synovium from OA patients, there is a high number of colocalizations in other joint diseases of autoimmune nature which are etiologically fundamentally different from OA (RA, JIA and AS). This is interesting, and perhaps not surprising, but there is no detailed analysis of how many of these colocs are new or have been seen by profiling individual immune cells and what is specific for their synovium dataset. The developed method is interesting, but needs an independent validation. The biology interpretations are profoundly lacking.

Response:

Thank you for your constructive comments! We would like to address these comments from the following four parts:

1) There are three key points of the novelty of our work:

- First, we identified 5,381 independent eQTLs and 4,765 eQTL genes based on the genomic and transcriptomic features of human synovium in up to 245 OA patients. With much larger sample size than the previous study (245 vs 77), our work provides an eQTL resource for understanding the role of synovium in arthritis.
- Second, by integrating our identified synovium eQTLs with GWAS summary data for multiple arthritis diseases, we uncovered many novel effect genes which have not been reported before.
- Third, we developed the eQTac method, which could identify variants that could affect gene expression by affecting chromatin accessibility without population scale ATAC-seq data.

We have added the above statement in the introduction section to make sure our

novelty is clear (Line 69-79).

2) For the colocalized genes, we compared our results with previous studies using eQTL data from GTEx or immune cells to identify colocalized genes. Details of these studies are listed in Supplementary Table 7. For AS and JIA, since the original GWAS study didn't perform colocalization analysis with eQTL data from the GTEx and BLUEPRINT project, we also performed colocalization analysis using these two datasets for comparison.

As shown in Figure 4b, the numbers of novel genes specifically identified by our synovium dataset for OA, RA, and AS were 18, 18, and 2, respectively. As for JIA, the only one colocalized gene was also identified by previous studies. We performed PPI analysis for the 84 genes using the STRING database[1]. As shown in Supplementary Fig. 10, the hub genes with the highest degree in the network (*CCR6*, *CD40*, *IRF5*, *ERBB2*, and *LRRK2*) (Supplementary Fig. 10) mainly participate in the immune-related process. *CCR6*, *CD40*, and *IRF5*, are well-known autoimmune disease genes associated with RA or AS[2-5]. Specifically, *ERBB2* and *LRRK2* are firstly identified by our synovium dataset as colocalized genes for RA and AS, respectively. *ERBB2* is a known oncogene with significant role in mediating tumor immune response[6, 7]. *LRRK2* is highly expressed in immune cells[8-10]. Mutant *LRRK2* could exacerbate immune response and neurodegeneration in a chronic model of experimental colitis[11]. Moreover, most of the novel colocalized genes identified by our synovium dataset formed interactions with known genes, suggesting that these genes might be closely related to influence the development of diseases together. Future functional studies are encouraged to explore their potential mechanisms. The corresponding descriptions have been added to the results section (Line 209-225).

3) We validated our eQTac method in another independent dataset (PMID: 29988122) as suggested and the results supported the robustness of our method (AUC = 0.81). The analysis details are described in the response to your last comment.

4) To make sure the biology interpretations of our study are appropriate, we have

extended the interpretation of our key findings in the results and discussion section (Line 210-226, 336-387).

Reference

1. Szklarczyk, D., et al., *The STRING database in 2023: protein-protein association networks and functional enrichment analyses for any sequenced genome of interest*. Nucleic Acids Res, 2023. **51**(D1): p. D638-D646.
2. Tawaraishi, T., et al., *Identification of a novel series of potent and selective CCR6 inhibitors as biological probes*. Bioorg Med Chem Lett, 2018. **28**(18): p. 3067-3072.
3. Grewal, I.S. and R.A. Flavell, *A central role of CD40 ligand in the regulation of CD4+ T-cell responses*. Immunol Today, 1996. **17**(9): p. 410-4.
4. Marzaioli, V., et al., *CD209/CD14(+) Dendritic Cells Characterization in Rheumatoid and Psoriatic Arthritis Patients: Activation, Synovial Infiltration, and Therapeutic Targeting*. Front Immunol, 2021. **12**: p. 722349.
5. Manni, M., et al., *Regulation of age-associated B cells by IRF5 in systemic autoimmunity*. Nat Immunol, 2018. **19**(4): p. 407-419.
6. Mei, J., et al., *Clinical and molecular immune characterization of ERBB2 in glioma*. Int Immunopharmacol, 2021. **94**: p. 107499.
7. Wu, S., et al., *HER2 recruits AKT1 to disrupt STING signalling and suppress antiviral defence and antitumour immunity*. Nat Cell Biol, 2019. **21**(8): p. 1027-1040.
8. Oun, A., et al., *LRRK2 protects immune cells against erastin-induced ferroptosis*. Neurobiol Dis, 2022. **175**: p. 105917.
9. Wallings, R.L. and M.G. Tansey, *LRRK2 regulation of immune-pathways and inflammatory disease*. Biochem Soc Trans, 2019. **47**(6): p. 1581-1595.
10. Tian, Y., et al., *LRRK2 plays essential roles in maintaining lung homeostasis and preventing the development of pulmonary fibrosis*. Proc Natl Acad Sci U S A, 2021. **118**(35).
11. Cabezudo, D., et al., *Mutant LRRK2 exacerbates immune response and neurodegeneration in a chronic model of experimental colitis*. Acta Neuropathol, 2023. **146**(2): p. 245-261.

Other concerns:

Comments 1:

1. I am a bit confused by the fact that in Figure 1 authors categorize continuous variables. In my opinion it either should be continuous or if distances and effect sizes are converted to categories those categories should be explained. Distance (near and far) and effect size (low and high) are extremely ambiguous. In 1C and 1D would it be better to perform a simple linear regression of two continuous variables? What is the meaning of comparing the two neighbouring violon plots?

Response: Thank you for the comment!

We also performed linear regression analysis. As shown in Supplementary Fig. 3 and Supplementary Fig. 4, the correlation between distance and effect size was Pearson $r = -0.14$, $P = 2.84 \times 10^{-23}$, and the correlation between effect and CVs was Pearson $r = 0.22$, $P = 7.32 \times 10^{-53}$. We have included this results in the main text (Line 107-112).

In Figure 1c-d, we categorize the distance/effect size into different quantiles mainly for better illustration. Due to the large number of data points and their non-normal distribution, it is hard to see the correlation trend from direct scatter plot (Supplementary Fig. 3 and Supplementary Fig. 4). In the revised Figure 1c and 1d, we have added a fitting line to the median values to better illustrate the correlation. To be clear, we have removed the ambiguous terms: “near and far”, “low and high”. We revised the Figure 1 legend to be clear: larger quantile number indicates larger distance/effect size values.

Comments 2:

2. I am failing to grasp the significance of Figure 3 and ATAC-seq analysis. It is well accepted that eQTLs need to be in active/open chromatin in order to express their effect on gene expression. What are the conclusions here and how this differs to any other eQTLs?

Response:

Thank you for the comment. We agree with you that the ATAC-seq analysis result is similar to other eQTLs. We have moved this result to Supplementary Fig. 6.

Instead, the current Figure 3 emphasized that eQTL variants are enriched in transcription factor (TF) binding sites in open chromatin regions (line 165-171). The top 10 TFs are enriched in immune-related gene ontology (GO) terms (Figure 3c, supplementary Table 4). Specifically, 7 of the 10 TFs have been reported to be involved in the immune cell differentiation or immune response process.

Comments 3:

3. The sharing between synovium eQTLs and GTEX tissues is a bit unexpected. What would be the explanation for having more shared eQTLs with nerves and thyroid than with the kidney or brain? Have you tried more stringent methods (for example mashR

PMID: 30478440)? Since you see many immune signatures why didn't you compare them with many datasets coming from immune cells (e.g. DICE, BLUEPRINT, OneK1k ...)?

Response:

Thank you for the comments.

1) We noticed that our previous results using the pi0 method might be affected by the sample size of the eQTL data. As shown in the following Figure A, the pi0 sharing scores are positively associated with the sample size ($R^2 = 0.82$, $P = 2.58 \times 10^{-19}$). Therefore, more shared eQTLs with nerves and thyroid than with the kidney or brain might mainly be due to the difference of their sample sizes (thyroid: 574, tibial nerves: 532, kidney: 73, brain: 114-209).

2) As you suggested, we used mashR instead to analyze the sharing between synovium and GTEx tissues. This method has the advantage of improving effect estimates and allowing for more quantitative assessments of effect-size heterogeneity. It is indeed more stringent, and a much weaker correlation was observed between the sharing score and sample size (following Figure B, $R^2 = 0.1$, $P = 0.024$). The hierarchical clustering results (Supplementary Fig. 8) showed that fibroblast and muscle are most similar to synovium, which is consistent with the fact that there are over 70% synovial fibroblast in synovial tissue[12]. We also noticed that synovium shared about 57.5% eQTLs with the whole blood (Figure 3d).

Figure. The scatter plot of sample sizes from 49 GTEx tissues and the corresponding sharing score with synovium eQTL from the pi0 method (A) and mashR method (B).

3) As you suggested, we further compared our synovium eQTL data with immune cell eQTLs from DICE and BLUEPRINT (data from OneK were not used since the beta and standard error values required by mashR are not available). The results (Supplementary Fig. 9) showed that our synovium shared over 40% with stimulated CD4+ and CD8+ T cells, NK cells and B cells, supporting the immune signatures we observed. The score is higher than that obtained using the previous synovium eQTLs data[13]. We have included these results in the revised manuscript (Line 181-187).

Reference

12. Chou, C.H., et al., *Synovial cell cross-talk with cartilage plays a major role in the pathogenesis of osteoarthritis*. *Sci Rep*, 2020. **10**(1): p. 10868.
13. Steinberg, J., et al., *A molecular quantitative trait locus map for osteoarthritis*. *Nat Commun*, 2021. **12**(1): p. 1309.

Comments 4:

4. The most novel part is the development of eQTac. Although it is very intriguing it needs an independent validation to make sure that these signals are real. There are several datasets that have eQTLs and ATAC-seq QTLs (e.g. PMID: 35591976; PMID: 29988122). In my opinion, authors should validate their method by comparing the data to the ground truth eQTac signals.

Response:

Thank you for the comments. As you suggested, we have applied the dataset you mentioned[14] (PMID: 29988122), which contains genotype, RNA-seq, and ATAC-seq data from 92 individuals to validate our method. The dataset from another study (PMID: 35591976) was not used because our application for data access was not approved.

For the genotype data, we performed imputation analysis and then SNPs with MAF < 0.1 were removed due to the small sample size. For the ATAC-seq data, we used macs2 to call the peaks and extract the 100bp sequences around the summits of peaks as the positive training set. The gene expression data were processed using the same pipeline as our eQTL calculation. The actual open chromatin scores of PRE regions were defined as the inverse normalized transformed reads counts from ATAC-seq data. Significant eQTL genes was subjected to subsequent analysis. The correlation results

using linear regression analysis for the open chromatin scores and target gene expression values were used to define the ground truth. That's, PRE-gene pairs with $P < 0.05$ was defined as true correlation. Then we used our eQTac method to identify the significant PRE-gene pairs. As shown in Figure 5b, the area under the ROC (receiver operating characteristic) curve (AUC) is 0.81, supporting the robustness of our method. We have included these results in the revised manuscript (Line 288-292, 795-808).

Reference

14. Gate, R.E., et al., *Genetic determinants of co-accessible chromatin regions in activated T cells across humans*. Nat Genet, 2018. **50**(8): p. 1140-1150.

REVIEWERS' COMMENTS

Reviewer #1 (Remarks to the Author):

The authors have largely addressed my concerns. I have no further comments.

Reviewer #2 (Remarks to the Author):

The authors have satisfactorily address the concerns on the previous version of the manuscript.

Reviewer #4 (Remarks to the Author):

The reviewer's comments have been addressed. The authors have done great work to further strengthen their manuscript. In particular, I appreciate the authors rigorously demonstrating eQTac performance